# Green Metallic Nanoparticles for Cancer Therapy: Evaluation Models and Cancer Applications

**DOI:** 10.3390/pharmaceutics13101719

**Published:** 2021-10-18

**Authors:** Ernesto Tinajero-Díaz, Daniela Salado-Leza, Carmen Gonzalez, Moisés Martínez Velázquez, Zaira López, Jorge Bravo-Madrigal, Peter Knauth, Flor Y. Flores-Hernández, Sara Elisa Herrera-Rodríguez, Rosa E. Navarro, Alejandro Cabrera-Wrooman, Edgar Krötzsch, Zaira Y. García Carvajal, Rodolfo Hernández-Gutiérrez

**Affiliations:** 1Departament d’Enginyeria Química, Universitat Politècnica de Catalunya, ETSEIB, Diagonal 647, 08028 Barcelona, Spain; ernesto.tinajero@upc.edu; 2Biotecnología Médica y Farmacéutica, Centro de Investigación y Asistencia en Tecnología y Diseño del Estado de Jalisco, A.C., Av. Normalistas 800, Col. Colinas de La Normal, Guadalajara 44270, Mexico; mmartinez@ciatej.mx (M.M.V.); jbravo@ciatej.mx (J.B.-M.); fflores@ciatej.mx (F.Y.F.-H.); sherrera@ciatej.mx (S.E.H.-R.); 3Facultad de Ciencias Químicas, Universidad Autónoma de San Luis Potosí, Av. Dr. Manuel Nava, Zona Universitaria, San Luis Potosí 78210, Mexico; daniela.salado@uaslp.mx (D.S.-L.); gonzalez.castillocarmen@uaslp.mx (C.G.); 4Cátedras CONACyT, México City 03940, Mexico; 5Centro Universitario de la Ciénega, Cell Biology Laboratory, Universidad de Guadalajara, Av. Universidad 1115, Ocotlán 47810, Mexico; zlopez@gmx.net (Z.L.); knauth@gmx.de (P.K.); 6Departamento de Biología Celular y Desarrollo, Instituto de Fisiología Celular, Universidad Nacional Autónoma de México, Circuito Exterior s/n, Ciudad Universitaria, México City 04510, Mexico; rnavarro@ifc.unam.mx; 7Centro Nacional de Investigación y Atención de Quemados, Laboratory of Connective Tissue, Instituto Nacional de Rehabilitación “Luis Guillermo Ibarra Ibarra”, México City 14389, Mexico; aca_w@yahoo.com.mx (A.C.-W.); kroted@yahoo.com.mx (E.K.)

**Keywords:** nanomaterials, metal-based nanoparticles, biosynthesis, cancer treatment, in silico, in vitro, ex vivo, in vivo

## Abstract

Metal-based nanoparticles are widely used to deliver bioactive molecules and drugs to improve cancer therapy. Several research works have highlighted the synthesis of gold and silver nanoparticles by green chemistry, using biological entities to minimize the use of solvents and control their physicochemical and biological properties. Recent advances in evaluating the anticancer effect of green biogenic Au and Ag nanoparticles are mainly focused on the use of conventional 2D cell culture and in vivo murine models that allow determination of the half-maximal inhibitory concentration, a critical parameter to move forward clinical trials. However, the interaction between nanoparticles and the tumor microenvironment is not yet fully understood. Therefore, it is necessary to develop more human-like evaluation models or to improve the existing ones for a better understanding of the molecular bases of cancer. This review provides recent advances in biosynthesized Au and Ag nanoparticles for seven of the most common and relevant cancers and their biological assessment. In addition, it provides a general idea of the in silico, in vitro, ex vivo, and in vivo models used for the anticancer evaluation of green biogenic metal-based nanoparticles.

## 1. Introduction

Despite the significant progress in studying, understanding, and treating cancer, the number of cases continues to rise, thus remaining a leading cause of death worldwide. Early diagnosis and screening are key aspects in the battle against cancer, which allow increases in therapies’ efficiency and reductions in cancer death rates. There are different modalities to treat cancer: surgery, chemotherapy [1], immunotherapy, phototherapy [2], radiation therapy [3], and targeted therapy [4], among others. Although conventional chemotherapy has allowed us to treat several types of cancer, there still are drawbacks in its use, such as poor bioavailability, high dose requirements, multidrug resistance, side effects, and lack of selective and specific targeting to tumor tissue [5]. Instead, targeted therapy uses therapeutic and antineoplastic agents capable of targeting specific genes and proteins involved in the growth and survival of tumor-forming cancer cells [4]. Their administration to solid tumors is carried out through the bloodstream, generally using injections of micro and nanoparticles [6]. Specific targeting of cytotoxic drugs to malignant cells using green biogenic gold (AuNPs) and silver-based nanoparticles (AgNPs) appears to be a promising strategy, but until now it has not been well developed.

The nanotechnology applied to cancer diagnosis and treatment emerged as a promising alternative with the potential of successfully combining the advances made at the nanoscale with cellular and molecular components that may allow overcoming the physiological and technological limits of conventional cancer treatment modalities. This approach has facilitated the coupling of nanoparticles (NPs) synthesized via the green biological pathway with target molecules, allowing an efficient interaction with biological systems [7]. In most cases, NPs are part of a complex multishell cancer cell targeting delivery system that serves as antineoplastic drug nanocarrier [8]. However, NPs alone are able to work as cytotoxic agents because, due to their physicochemical properties, they tend to accumulate passively into the tumoral tissue [1,2]. Cytotoxic agents (drugs, encapsulated drugs, or NPs) can be released from the carrier to the bloodstream or from the carrier directly into the tumor. In both cases, adsorption of plasma and interactions with the surrounding tissues occur until their metabolization and clarification [2]. As expected, NPs must cross several physiological barriers to reach cancer cells. Many physical, chemical, and biological phenomena are involved during the NPs and physiological barrier encountering, e.g., interactions with proteins, cells, and the dynamics of NPs from the blood to the tumor, etc., which significantly modify the behavior and effect of NPs [3].

There are many considerations to contemplate when formulating NPs as carriers or cytotoxic agents [1]. The controlled release of drugs in organs, tissues, or cells does not behave under the laws of diffusion of Fick, so it is necessary to develop intelligent delivery platforms that respond to multiple stimuli [2]. Green Au and AgNPs are promising antitumor agents gaining specificity towards the cancer cells **[4]**. These NPs could act as nanocarriers to passively target the tumor via the enhanced permeability and retention effect (EPR) or to actively target solid tumors via a ligand–receptor interaction [6]. NPs stability modulates their biological response [9] modifying their cellular uptake, bioavailability, and toxicity during their evaluation using in vitro, ex vivo, and in vivo models [10,11,12]. The main goal is to improve therapeutic index of current cancer therapies with higher specificity, sensitivity, and efficacy using green biogenic AuNPs and AgNPs [13]. Indeed, AuNPs are the most promising platform to diagnose and treat tumors. Its easy surface modification, unique optical properties, high biocompatibility, and small size favor extravasation and access to the tumor microenvironment [14,15]. AuNPs have been mainly engineered to serve as targeting delivery vehicles, molecular probes, and biosensors [16]. Moreover, AgNPs are well known for their antibacterial activity. The resistance to cationic silver (Ag^+^) has been recognized for many years, but it has recently been found that bacterial resistance to AgNPs is also possible [17]. AgNPs have also been demonstrated to be useful in the treatment of multiple types of cancer [18]. The anticancer properties of AgNPs are carried out by various mechanisms, for example, the release of silver ions and the formation of intracellular free radicals, leading to deregulation of critical cellular mechanisms, cell damage, and death [18]. These properties of AuNPs and AgNPs are interesting and important because they could allow us to solve the drawbacks that occur with chemotherapy, such as poor bioavailability, side and adverse effects due to high doses, non-specific targeting, and the development of resistance to multiple drugs [5].

This review presents the state-of-the-art progress of AuNPs and AgNPs obtained by green biological methods and their potential application in cancer therapy. An overview will be given on the biological routes used to obtain NPs, on the methodologies used to modify their surface and develop stimulus-responsive smart nanosystems, on the physicochemical aspects that impact their behavior in biological systems, and on several models used to evaluate the antitumor properties of these nanoparticles. Moreover, it will address how NPs take advantage of the tumor microenvironment to target cancer cells, focusing on therapies for six types of cancer. Finally, the review describes future perspectives associated with these techniques as effective strategies to enrich current clinical trends.

## 2. Overview: Green Biosynthesis of Metallic Nanoparticles

### 2.1. Nanomaterials Classification: From Geometrical Aspects to Functionality

Nanotechnology covers the applicability of a wide variety of materials with dimensions ranging, according to the International Organization for Standardization (ISO), from one to one hundred nanometers (1 nm = 10^−9^ m) [19]. It co-exists with other scientific and technological fields derived from chemistry, physics, and biology, which allow us to produce and evaluate this nanosized atom confinement’s unique and enhanced properties.

Typically, nanomaterials (NMs) are classified based on their size, shape, chemical composition, and other tunable characteristics that define their specific properties and functionalities. Regarding their size or dimensionality, there are zero-dimensional (0D) dots, clusters, and particles; one-dimensional (1D) fibers, wires, and rods; two-dimensional (2D) films, plates, and networks; and three-dimensional (3D) nanocomposites with, e.g., embedded clusters or equiaxed crystallites [20]. The sphericity, flatness, and aspect ratio characterize the morphology of NMs. A low aspect ratio identifies the shape of a spherical, cubic, prism, or helical nanoparticles, while a big aspect ratio describes zigzag, helice, and belt morphologies. From the chemical composition point of view, a common classification groups NMs into (i) carbon-based, (ii) organic, (iii) inorganic, and (iv) hybrid NMs, the latter being associated with more than one quality to build synergistic multi-modal agents via intra- or intermolecular forces.

Carbon-based NMs are allotropes with sp^2^ carbon atoms that show remarkable physicochemical properties for variable sectoral applications. Organic NMs, such as liposomes, dendrimers, micelles, or polymers are mainly composed of molecular carbon, hydrogen, oxygen, nitrogen, sulfur, and phosphorus. Inorganic NMs, those with particular intrinsic optical, magnetic, electronic, mechanical, or thermal properties are a broad group of materials consisting of specific s, p, d, and f-block elements to form single- or multi-element NMs [21]. Thus, these NMs may contain alkali metals (Li, K, and Cs), alkaline earth metals (Mg, Ca, Sr, and Ba), transition metals (Ti, V, Cr, Mn, Fe, Co, Ni, Cu, Y, Zr, Mo, Tc, Ru, Ph, Pd, Ag, W, Ir, Pt, and Au), post-transition metals (Al, Zn, Cd, In, Sn, Hg, Pb, and Bi), metalloids (Si, As, Sb, and Te), non-metals (H, C, N, O, P, S, and Se), lanthanides (La, Ce, Pr, Nd, Sm, Eu, Gd, and Yb), or an actinide (Ac) [21,22]. Finally, the hybrid NMs can be sectioned as organic-on-inorganic, i.e., organic matter used to modify inorganic NMs, or vice versa, inorganic-in-organic, using inorganic constituents to modify organic materials. This group is particularly attractive for developing enhanced nanoagents for emerging biomedical applications (sensing, diagnosis, and therapy).

Based on the origin, i.e., their source of generation, NMs are further classified as natural or synthetic. Natural NMs can be formed by earth/spatial phenomena, biological species, or incidental human activities. Synthetic NMs are obtained following physical, chemical, biological, or mixed procedures that (i) use macroscopic materials for their miniaturization (top-down) or (ii) use basic molecular or atomic units to build up nanostructures (bottom-up). These approaches allow the control of NM properties for a more convenient large-scale production.

A remaining challenge in the development of synthetic NMs for innovation is the safe-by-design. This concept must be regularly implemented to evaluate the life cycle and final disposition of NMs [23]. At the same time, the use of effective and sustainable methods that assure the 12 principles of green chemistry [24], e.g., by applying biological synthesis, or greener, simple, and reproducible assisted physicochemical procedures, is still challenging. Hence, safe NMs can be created to play functional activities to benefit living beings, including humans, and the environment. In emerging medical fields, which combine conventional protocols with nano-technological tools, NMs can serve as selective agents to deliver, among others: drugs, vitamins, hormones, and amino acids via their hybridization, surface chemistry, and bioconjugation. NMs can also be used as radioenhancers or contrast agents in the perspective of theranostics [25]. In particular, the principles of the biological synthesis routes, using plants and microorganisms, are described below to introduce this green approach in developing metallic nanoparticles that are currently evaluated to improve cancer therapy. The preparation of metal-based nanoparticles (MNPs) using green approaches provides NPs with well-defined sizes, morphologies, and stability. The factors that govern the stability of MNPs can be described based on aggregation state, composition, and surface properties. Two crucial steps occur during the synthesis of MNP: the reduction and stabilization (capping) of NPs. The combinations of molecules found in the extracts of plants or microorganisms act as reducing agents and stabilizers, which promote the formation of nanoparticles and inhibit their agglomeration, thus favoring their stability [9].

### 2.2. Green Biosynthesis of MNPs

MNPs are promising agents that combine optic, magnetic, electric, and thermal properties in a single smart (stimuli-responsive) nanosystem to enhance cancer treatment. This type of MNPs can be synthesized following different bottom-up routes, the biological strategies being very advantageous as renewable resources and low-cost methods and equipment are required this is especially important for poor (“third word) countries to improve their anticancer repeertorie. In addition, biosynthesis is a straightforward eco-friendly approach capable of producing non-toxic and biodegradable coatings and nuclei that may positively impact cells and organisms. In this context, derived from plant tissues (seed, root, stem, leaves, flowers, fruit, and peel) and microorganisms (fungi, bacteria, yeast, and algae) are used as reducing agents and biocompatible coatings in the development of potential metallic nano-therapeutics. Figure 1 schematizes the biosynthesis of MNPs showing the species used to extract the active compounds from plants (phytochemicals) and microorganisms (e.g., enzymes), which serve as reducing and capping agents. In this sense, reaction parameters, such as pH, temperature, time, and concentration must be controlled to obtain well-dispersed, homogeneous, and stable nanoparticles.

#### 2.2.1. Phytochemical Synthesis

In recent years, the phyto- (from the greek phytón, meaning “plant”) chemical production of metallic nanoparticles has attracted enormous attention due to its simplicity and sustainable implementation. Phytochemicals are plant secondary metabolites classified based on their chemical structure into carbohydrates, lipids, phenolic acids, terpenoids, alkaloids, and other nitrogen-containing compounds [26]. They are extracted from plant tissues and purified for specific nano-engineering tasks. Briefly, multiple phytochemicals or purified compounds can be added, in particular ratios, to aqueous solutions containing metallic salts (metallic mono- or polyvalent ions) and incubated at light-controlled conditions to reduce the precursors and form zero-valent MNPs. Several natural extracts have already been tested to prepare AuNPs and AgNPs with variable size and shape and used for antibacterial and anticancer applications [27]. Additionally, plant extracts have, among others, antioxidant, antifungal, anti-inflammatory, antiviral, and antibacterial properties, making them excellent candidates to modify the surface of inorganic NPs. Hence, phytochemicals act as reducing, capping, and bioactive molecules.

For instance, Ranoszek-Soliwoda et al. [28] reported the synthesis of monodispersed AgNPs using plant extracts rich in flavonoids and polyphenols, which provide higher reducing power. The AgNPs were obtained by adding cacao bean and grave seed extracts to silver nitrate in the absence and presence of sodium citrate. The authors found that sodium citrate plays a fundamental role in homogenizing the size and shape of the bio-derived AgNPs, suggesting a complexation process for the accurate bio-conjugation. Boomi et al. [29] used leaf extracts from the *Croton sparsiflorus,* containing carbonyl, phenolic, and amino groups capable of reducing auric ions to form AuNPs. These NPs presented, among other biomedical properties, anticancer activity against human liver cancer cells (HepG2).

#### 2.2.2. Microorganism-Assisted Synthesis

Microorganism-assisted synthesis is another so called environmentally friendly approach to obtain MNPs for biomedical applications. Microorganisms, such as bacteria, fungi, yeasts, and microalgae, produce inorganic NMs due to naturally detoxifying processes. In particular, the microorganisms’ intra- or extracellular redox potentials, characterized by leakage of enzymes, metabolites, and proteins, are critical parameters in reducing metal precursors.

For instance, Patil et al. [30] evaluated the extracellular biosynthesis of AuNPs using *Paracoccus haeundaensis* BC74171T, a marine bacterium. The cell culture supernatant was treated to obtain these NPs. Authors obtained spherical AuNPs with an average diameter of 20 nm, which are non-toxic for normal cells but showed an anti-proliferative effect in cancer cells. Even though microorganisms have enormous capabilities of producing NMs under mild conditions, i.e., without the necessity of adding reducing agents or surfactants as their efflux pumps make available what is required, bacteria’s surface affinity is essential to detonate the intracellular synthesis. Therefore, bacteria-assisted synthesis is time consuming and shows lower yields compared to other green methods [27]. Additionally, it has been reported that particular fungal enzymes can modulate the shape of NMs. On the other hand, viruses, which are considered non-living catalysts, can be used in host cells to generate bio-nanotechnological tools. Few studies have shown the production of NMs via viruses action [31].

### 2.3. Development of Stimuli-Responsive MNPs

MNPs can remodel the tumor microenvironment (TME) by changing unfavorable conditions into therapeutically accessible ones, which may be reached by applying external stimuli, such as light, heat, ultrasonic radiation, or magnetic fields. Hence, those stimuli may change the redox potential of biological systems and generate reactive oxygen species (ROS) that sensitize tissues. Stimuli-triggered cancer therapeutics depends on the modulation of the NMs properties, from the tumor outside or inside. pH, redox potential, and hypoxia are the internal conditions that lead to tumor sensibilization. The combination of MNPs with external and internal stimuli can trigger the on-demand release of therapeutic molecules, which augments the therapeutic efficacy of anticancer therapies [1,2,8]. The surface functionalization of MNPs with organic molecules or macromolecules is an excellent tool to stabilize the NP and further manipulate its properties. From a chemistry point of view, mercapto (R-SH) derivatives are widely used as ligands for NPs surface modification due to the strong affinity of thiol moieties to transition metal surfaces, which form (polar) metal-thiolate linkages [32].

Here, we highlight two effective strategies for introducing functional groups. Figure 2 shows: (i) the first method that consists of introducing the functional ligand in a single step and (ii) the second method that involves adding an intermediate compound to firstly reacts onto the metal surface. To perform this, bifunctional organic compounds are used. Thus, one reactive group is attached to the NP surface and the other may be the final functionality or an anchoring group that is conjugated in the second step [5]. It is noteworthy to point out that the fewer the steps taken to conjugate certain functionality onto a metal surface, the shorter the time invested, the more affordable its fabrication, and the higher the yields.

Gold and silver-based NPs are of great interest in medicine. MNPs can be functionalized with a variety of functional groups [33]. This review mainly discusses the development of MNPs functionalized with macromolecules (polymers) or small molecules to render smart nanosystems. Figure 3 collects a library of compounds commonly used to decorate the surface of MNPs. Other compounds are instrumental for surface functionalization (e.g., nucleic acids, antibodies protein, lipids, or peptides) but are outside the scope of this review. We refer the interested readers to recent and excellent reviews that discuss their use in great detail [34,35,36].

#### 2.3.1. Thermo-Responsive Polymers

Temperature is a well-controlled stimulus that can be easily applied and removed through thermal heating or photo-illumination to trigger a specific therapeutic effect [37]. Thermo-responsive polymers are ideal when the stimulus needs to be applied artificially rather than exploiting peculiar conditions of a targeted tissue or organ. Figure 3 displays some thermo-responsive polymers that have been used to coat MNPs.

A thermo-responsive polymer exhibits a phase transition temperature that corresponds to that at which a drastic change in solubility occurs. Polymers presenting a low critical solution temperature (LCST) are more often chosen for biomedical purposes. The LCST can be adjusted to achieve the desired temperature above the body temperature so that the drug is administered as soon as it reaches the specific LCST [38]. For instance, thermo-responsive AuNPs consist of a gold core surrounded by a thermo-responsive polymer shell that can be either covalently linked or physically adsorbed onto the surface.

Iida et al. [39] reported the synthesis of 3–10 nm thermo-responsive AuNPs coated with a self-assembled monolayer of thiol-terminated poly(ethylene glycol) (PEG, n = 6) ligands with ethyl, iso-propyl, and propyl alkyl heads. A ligand exchange reaction was carried out to coat the AuNPs, which displayed an assembly and disassembly in response to the temperature change in an aqueous solution. Even if the authors did not report a proof of concept, i.e., the entrapping and controlled release of a model drug, these thermo-responsive AuNPs can be considered as potential smart delivery systems for cancer therapy. On the other hand, poly(2-alkyl-2-oxazoline) (PAOx) derivatives have been utilized to surrogate PEG as their thermal-responsiveness can be modified by varying the alkyl substituent at the 2-oxazoline position [40,41]. Poly(N-alkyl-acrylamide) (PNIPAM) derivatives are widely used for engineering thermo-responsive nanosystems as they show a LCST around 32 °C Li et al. [42] prepared thiol-terminated PNIPAM and a PEG-*b*-PNIPAM diblock copolymer to modify AuNPs by forming Au-S bonds. Interestingly, the authors observed that the LCST could be easily adjusted by increasing the PEG chain segment. Poly(*N*-vinyl caprolactam) (PNVCL) is, together with PNIPAM, one of the most popular temperature-responsive polymers. PNVCL exhibits a LCST behavior in water between 30 and 32 °C. The main role of PNVCL is to solubilize/stabilize NMs, providing them a distinct temperature sensitivity [43,44]. Liu et al. [45] described core-corona gold/poly(vinyl alcohol)-*b*-poly(*N*-vinyl caprolactam) (gold@PVOH-*b*-PNVCL) NPs fabricated via the in situ reduction of HAuCl_4_ using NaBH_4_ and in the presence of PVOH-*b*-PNVCL copolymers. Upon temperature variations, the gold@PVOH-*b*-PNVCL NPs showed a thermo-induced phase transition and a good reversibility. In addition, this nanosystem could encapsulate a significant amount of Nadolol (a drug used to treat high blood pressure), the fast release of which was detected above the LCST.

#### 2.3.2. pH-Responsive Mechanisms

It has been documented that the pH of tumor cells and TME is a relevant factor in the study and development of effective therapies against cancer. Regularly, cancer cells have a slightly more acidic extracellular pH than normal cells, with differences varying from 0.3 to 0.7 pH units. [46]. The engineering of pH-responsive systems has arisen as the pH differences between normal and pathological tissue encouraged the utilization of local pH as a natural trigger for drug-controlled release. Two main strategies are involved for engineering pH-responsive MNPs: (i) the use of organic molecules or polymers that possess, in their chemical structure or repeating units, ionizable chemical groups, such as amines, phosphoric acids, and carboxylic acids (protonation/deprotonation process) and (ii) the use of molecules bearing pH-sensitive bonds that are cleaved under acidic conditions (bond cleavage) [47].

pH-Sensitive polymers are polyelectrolytes with ionizable groups in their backbone or in either side or end groups. Figure 3 displays a selection of polymers and organic molecules that hold ionizable groups and that are commonly used to modify the surface of MNPs. Smart polymers are ionized when the pH and ionic composition of an aqueous medium shift, dramatically changing their conformation. Polymers undergo pH-sensitive conformational changes in three different ways: a) dissociation, b) destabilization (via collapse or swelling), and c) changes or partition coefficient between the drug and the vehicle. Among the most commonly used pH-sensitive polymers for decorating AuNPs, we encounter anionic polymers, such as poly(acrylic acid) (PAA) and poly(methyl methacrylate) (PMMA) [48,49], and cationic polymers, such as chitosan (CS) [50] and poly(4-vinyl pyridine) (PVP) [51]. For instance, commercially available organic molecules have been utilized to decorate and design pH-responsive nanosystems. Fan et al. [52] reported reversibly responsive nanovesicles (NVs) by self-assembly of AuNPs with the commercially available 4-mercaptobenzonic acid (4-MBA) and oleylamine (OL). These self-assembled NVs dissociated into individual NPs under alkaline conditions and reassembled into NVs when the solution was brought back to acidic conditions; this event is due to the deprotonation and protonation of the 4-MBA. As a proof of concept, Rhodamine B (RhB) was loaded in the water-soluble Au NVs, which rapidly responded to alkali and acid stimuli demonstrating their pH-responsive release capability. Although authors did not report in vitro or in vivo studies, such elegant nanosystem showed great potential in the controlled release of active molecules.

The use of pH-sensitive bonds containing molecules is another common strategy to obtain more precise nanotherapeutics, which, via bond cleavage under acidic conditions, allows targeting drug release within pathological regions [53]. For instance, pH-sensitive linkers (Table 1), such as acetal, hydrazine, and ester bonds, can be cleaved by decreasing pH, serving thus as promising smart ligands for designing pH-responsive NPs [54].

Zayed et al. [56] exploited the in vitro drug release of doxorubicin hydrochloride (DOX·HCl) chemically bonded to AuNPs using breast cancer cells. They conjugated DOX·HCl via a hydrazone bond to the carboxyl functional group of a pre-formed polymer (undecanethiol-polyethylene glycol hydrazide). The DOX-PEG-undecanethiol conjugate reacted over the metal surface of NPs to obtain AuNPs functionalized with the exposed DOX on their surface. As expected, up to 80% of the chemically bound DOX·HCl was released in acidic acetate buffer (pH 5) after 72 h. Indeed, these results demonstrated the higher hydrolytic impact of the acidic conditions on the pH-sensitive hydrazone linker that led to the release of this amount of DOX.

#### 2.3.3. Light-Responsive Groups

The major advantage of light-responsive nanosystems is their temporal and spatial controllability; the drug-releasing behavior can be precisely controlled by applying specific light at a specific position. Drug delivery systems can respond to ultraviolet (UV), visible, and near-infrared (NIR) light. In order to prepare light-responsive inorganic NPs, their surface needs to be modified with photo-responsive materials, which can be activated/deactivated as they are subjected to radiation of a specific wavelength [53,57,58]. Recurring chemical moieties that render material photoresponsiveness include photochromic or cleavable groups (Figure 3).

In a photochromic group, a single UV/visible photon induce sufficient energy to achieve photochemical reactions. This type of group can transit reversibly between two structures upon irradiation. Among the photochromic groups that have been used to modify the surface of MNPs, azobenzene (AB) [57,58], spiropyran (SP) [59], and coumarin (Coum) [60] present a reversible transformation between their isomers (photoisomerization), which results in polarity and hydrophobicity changes that can lead to the organization or disassembly of NPs [61,62]. He et al. [60] reported Coum-functionalized AuNPs capable of performing reversible self-assembly based on the photolysis of coumarin in response to light irradiation. The authors demonstrated that this nanosystem could be self-assembled from the red disperse state to the purple aggregate state by irradiation with *λ*= 365 nm and then transformed back to the original disassembly state using UV irradiation.

On the other hand, photocleaved reactions are often used to affect particles’ integrity in terms of hydrophilic–hydrophobic balance and the stability of polymers’ building blocks to cleave linkages between drug molecules and NPs or to remove moieties that prevent particle binding. There exists an appealing approach based on the concept of photocaging where a biologically active molecule (e.g., ligand or drug) is temporally inactivated by protecting it with a photocleavable group (‘photocaged’) [63]. This caged molecule releases its parent species actively only when its photosensitive protective group is cleaved by UV irradiation. The 2-nitrobenzyl group (ONB) is widely used in organic synthesis as a protecting group and cleavable linker due to its high photocleavage efficiency upon near-UV light irradiation [64]. Although this approach seems to be promising, few reports have been mentioned. A communication reported by Agasti et al. [65] elegantly used this strategy. They reported the use of AuNPs for the photo-controlled release of a caged anticancer drug, 5-fluorouracil (5-FU), by pairing the drug to the NP surface through a photoresponsive ONB linkage. In such a system, the particle was used to cage and transport the bioactive cargo for an effective release upon long-wavelength UV irradiation. Another photocleavable group used is the diazirine (DA) ring that undergoes specific nitrogen elimination by irradiation at 355–370 nm or even at 405 nm [66].

#### 2.3.4. Magnetic Targeting and Theragnostic

The main problem of drug delivery systems is the lack of tissue selectivity. Magnetic NPs can overcome this limitation by directing them to the target using an external magnetic field. However, the benefit of magnetic NPs is also limited by the toxicity of many magnetic materials. Therefore, the magnetic core can be coated with noble metals to gain biocompatibility (Figure 3) [67,68]. Gold and silver stand out since Au has low bioactivity, and Ag has been largely used in biomedical applications [69]. Thus, the pairing of a superparamagnetic core with an inert and safe metal coating enriches MNPs properties. Moreover, this coating provides a chemically active surface ready for functionalization and the engineering of smart nanodevices [70,71].

For example, iron oxide nanoparticles (IONPs) or Superparamagnetic iron oxide nanoparticles (SPIONs) and AuNPs are very attractive for developing unique systems with high potential in cancer theragnostic (diagnosis and treatment with a single nanoagent). Indeed, even at low concentrations, IONPs are good magnetic hyperthermia and magnetic resonance imaging (MRI) contrast agents, and AuNPs have unique properties involving photons absorption, which make them suitable for cancer photothermal treatment and X-ray computed tomography (CT). The synthesis of magnetic AuNPs involves the iron or iron oxide core synthesis and the subsequent gold coating. The gold shell can be formed directly or indirectly onto the magnetic core. In the direct methods, the Au shell is formed directly on the core surface, while in the indirect methods, a “glue material” is used between the core and the Au shell [72]. One of the most efficient and simple methods to functionalize magnetic NPs is the sequential growth of metallic components (e.g., Ag or Au) onto the surface of IONPs core in a one-pot reaction [73]. Iron-gold NPs can be synthesized by reducing HAuCl_4_ [74,75]. For example, Eyvazzadeh et al. [76] synthesized core-shell gold-coated IONPs (Au@IONPs, 33 nm) to be used as an MRI contrast agent and a light-responsive agent for cancer photothermal therapy (PTT). A photothermal treatment, applied to the KB human nasopharyngeal carcinoma cell line in the presence of this nanosystem, killed approximately 70% of the cells. Au@IONPs alone did not trigger significant cytotoxicity to KB cells. Thus, the authors demonstrated the potential utility of these Au@IONPs for cancer PTT.

### 2.4. The Influence of the Physicochemical Properties of MNPs

The behavior and performance of the MNPs developed for cancer therapy are affected by several parameters. The size, shape, and surface characteristics greatly influence their therapeutic efficiency and efficacy [77,78]. In this sense, noble metal NPs, especially gold and silver, are being extensively used due to their excellent compatibility with biological systems [79]. The green biosynthesis of MNPs has an exceptional degree of repeatability, which supports the controllable self-assembly of NPs, offers efficient target delivery platforms, and provides NPs with many new functions that may have great potential to satisfy advanced cancer therapy [80].

Fundamentally, nanotechnology is all about size, so one of the most interesting aspects is how the properties of NPs change with this parameter. Au and AgNPs are plasmonic nanoparticles which size and shape strongly impact their spectral response Their surface properties, such as composition, functionalization, and charge will affect size (promoting aggregation), solubility in aqueous solution, and the ability to penetrate cells [77]. In general, in aqueous solution, there is a tendency to form aggregates that are much larger than the primary size of NPs. The trend for MNPs to aggregate depends on several factors, including surface functionalization, nanoparticle concentration, pH, and ionic strength [81].

MNP surface charge is closely related to many biological performances, such as biodistribution, stability, cellular uptake, and cytotoxicity. The charge interaction between particles and cells is an essential basis for their biological effect [77]. Various synthetic methods can modify the surface properties using reducing and stabilizer (capping) agents. The most common strategy for stabilizing MNPs is the use of agents that can be adsorbed onto the NPs surface. Different types of stabilizers have been successfully used, including surfactants, small ligands, polymers, dendrimers, cyclodextrins, and polysaccharides, etc. They can induce subtle changes in NPs, favoring notable changes in their physicochemical and biological characteristics and impacting their therapeutic effects [79,82]. Furthermore, MNPs could be stabilized by achieving bulky groups (stearic strategy), such as organic polymers (chitosan). Additionally, their surface can be conjugated with biomolecules, such as DNA probes, peptides, or antibodies, used to target specific cells or components [83]. The capping agents stabilize the interface where NPs interact with their medium of preparation [82]. Green biosynthesis of Au and AgNPs involves natural protection agents for their stabilization [6]. Several eco-friendly processes for the rapid synthesis of MNPs have been reported using aqueous extracts of plant parts, such as the leaf, bark, roots, seed, flowers, fruits, and peel [84]. These stabilizing agents play a key role in altering the biological activities with an environmental perspective [82].

On the other hand, it is possible to synthesize Au and AgNPs in various forms, including spheres, rods, triangles, stars, rounds octahedral, prisms, and wires for a wide variety of applications [85,86]. For instance, spherical AuNPs with sizes ranging from 10 to 50 nm absorb light and show their particular plasmon peak around 520 nm. On the other hand, larger spheres have an increased scattering and peaks located at longer wavelengths (red-shifting). Big spheres scatter more photons than the small ones because they have larger optical cross-sections and because their albedo (ratio of change energy to total extinction) increases with size. When the shape of AuNPs change from spheres to rods, the surface plasmon resonance (SPR) band is split into two bands: (i) a strong band in the NIR region that corresponds to electron oscillations along the long axis, referred to the longitudinal band, and (ii) a weak band in the visible region at a wavelength similar to that of gold nanospheres [87,88]. In AgNPs, small spheres (10–50 nm) typically have a small absorbance peak near 400 nm, while larger spheres (100 nm) have a broader peak with a maximum that shifts toward longer wavelengths around 500 nm. Moreover, the spectra of larger spheres have a secondary peak at shorter wavelengths, resulting from quadrupole resonance and the primary dipole resonance [89]. The destabilization and formation of aggregates can lead to peak broadening or a secondary peak forming at longer wavelengths. Silver nanoprisms have a specific SPR ranging from 400 to 850 nm [86]. Stabilization of dispersive Au and AgNPs during a green biosynthesis course is essential. When green routes are used, the most commonly obtained shape is spheres [84,90]. For instance, Botteon et al. [91] reported the biosynthesis of AuNPs using Brazilian red propolis, a product of bees that exhibits anti-inflammatory, antitumor, antioxidant, and antimicrobial activities. The AuNPs average size was in the range of 8 and 15 nm and showed several geometrical forms, such as spheres, triangles, pentagons, hexagons, and rods. The cytotoxic activity of these biosynthetic AuNPs was evaluated in a human urologic cancer cell line. Oves et al. [92] reported the synthesis of AgNPs at room temperature using AgNO_3_ and the culture supernatant of *Stenotrophomonas maltophilia* strain OS4. The cuboid and homogenous obtained AgNPs showed a characteristic SPR at 428 nm with an average size of 93 nm.

Hence, this field of research has become a “hot” topic in recent years. However, the mechanism of NP formation (reduction and stabilization) and physicochemical properties must be well understood and experimentally validated. In addition, the use of Au and AgNPs as carriers to achieve passive or active targeting for cancer therapy can significantly improve the efficacy of the conventional anticancer drugs, reduce the death rate of cancer patients, and improve the quality of the patient’s life [93].

## 3. Cancer Evaluation Models

Traditionally, the safety and efficacy of anticancer drugs are evaluated through exploratory and confirmatory studies. The in vitro, ex vivo, and in silico studies are exploratory, while in vivo assays are classified as confirmatory [94]. Many questions remain regarding whether the exploratory results have any bearing on the effectiveness in the human body, which lead to develop additional and more appropriate methods for MNP evaluation. The safety and efficacy of Au and AgNPs are generally focused on determining conventional pharmacokinetic and pharmacodynamic parameters. As shown in Figure 4, several models have been developed to evaluate the biological performance of MNPs. The starting point for cancer research is usually two-dimensional (2D) cell cultures, i.e., the use of adherent standard and commercially available cell lines or primary cultures that are easily manipulated and rapidly generated. Three-dimensional (3D) cultures, such as organoids, are more accurate in vitro models, the main goal of which leads to mimic the TME [95,96]. Organoids are generated by deconstructing human tumors and culturing the tumor-derived cells in a semisolid extracellular matrix under well-defined conditions [96]. In most cases, the assays are conducted on organoids derived from the primary tumors, although a few examples of successful cultures derived from metastatic cancer sites exist. Ex vivo tumor models are generally the representation of the so-called explant models in which fresh tumors or fractions are surgically obtained and used for anticancer drug evaluation [97]. In vivo models mostly refer to mouse models and have been extensively reviewed elsewhere.

### 3.1. In Silico Analysis

Although in vitro, ex vivo, and in vivo experimental models are widely used to evaluate drugs and MNPs, significant challenges remain [98]. For example, NPs must cross several biophysical barriers to reach cancer tissue and fulfill their role [99]. Furthermore, the TME itself has its own barriers, which compromise its safety and efficacy. Understanding of the biochemical, biophysical, and mechanical processes taking place during the selective administration of NPs is complex since they depend on the dose and exposure time [98,99]. The penetration, distribution, accumulation, and targeting capacity of NPs are essential aspects that determine the effectiveness of anticancer agents. Generally, the prediction of these mechanisms is obtained from experimental models and, more specifically, at the in vivo preclinical and clinical stages. However, they might depend on the analyst’s perception causing failures and errors in predicting those interactions [1,98,99].

A valuable tool for understanding these interactions is the in silico approach [1]. Mathematical and computational modeling allows elucidating these processes that are often impossible or uneconomical. Kashkooli et al. suggest that mathematical modeling is an auspicious tool in optimizing the development of nano-scale formulations for the targeted delivery of anticancer agents, combining in silico and in vivo models to better understand their efficacy and safety [99]. Furthermore, if the mathematical models include variables related to the molecular pathways of cancer, genetics, and the possible mechanisms of interaction of NPs or drugs, personalized drugs and therapies could be developed [1].

### 3.2. Conventional 2D In Vitro Assays

In vitro 2D cell culture assays have the main objective of determining whether some treatments or substances cause toxicity, damage, apoptosis, or necrosis to a cell monolayer. Cell lines isolated from primary tumor tissues are commonly used. Green biosynthetic MNPs have gained importance in the last years because of their antimicrobial and/or anticancer properties. Cytotoxicity assays are generally quantitative and, by metabolizing specific substances with optical and redox properties, allow the evaluation of the effect of increasing concentrations of substances or NPs. These tests consider determining the half maximal inhibitory concentration (IC_50_), which represents a concentration capable of reducing the viability of the cell culture by 50% [100,101,102,103].

The most commonly used method to evaluate cell viability is the MTT test, which uses 3-(4,5-dimethylthiazol-2-yl)-2,5-diphenyltetrazolium bromide (MTT). The reduction of the tetrazolium salt indicates some general metabolic or enzymatic activity aspects that correspond to viable cells [104]. There are other methods based on the same principle, in which the salts are soluble in water. The MTS assay uses the 3-(4,5-dimethylthiazol-2-yl)-5-(3-carboxymethoxyphenyl)-2- (4-sulfophenyl)-2H-tetrazolium salt or soluble tetrazolium (WST), which has the advantage of no longer requiring the use of a solubilizer such as dimethyl sulfoxide (DMSO) [105,106,107]. Cell viability has also been determined with methods based on evaluating the integrity of the plasmatic membrane. One of these methods is based on the ability of viable cells to exclude the trypan blue dye (TB) [108,109]. The TB assay allows for direct identification and counting of live (unstained) and dead (blue) cells, which also has made it possible to determine the toxicity of MNPs on cancer cells [110,111].

Another recommended cell-based microscopic assay is the neutral red uptake (NRU) assay used to quantify metabolic activity. The neutral red uptake assay provides a quantitative estimation of the number of viable cells. It is based on the ability of viable cells to incorporate and bind the supravital dye neutral red in lysosomes [112]. Neutral red uptake has been employed to evaluate the toxicity induced by AgNPs synthesized by a green route in bladder (5637) and breast (MCF-7) cancer cell lines [113]. One more common method to determine cytotoxicity measures the activity of cytoplasmic enzymes released by damaged cells. Lactate dehydrogenase (LDH) is a stable cytoplasmic enzyme found in all cells. LDH is rapidly released into the cell culture supernatant when the plasma membrane is damaged. LDH activity can be quantified by using the NADH produced during the conversion of lactate to pyruvate to reduce a second compound in a coupled reaction [114]. This assay has been used to determine the ability of NPs to induce cytotoxicity in the breast cancer cell line MDA-MB-231 [102]. Most designs that seek to evaluate the effectiveness of new selective molecules or treatments against cancer include some controls, which, in most cases, comprise a cancer model cell line and a non-cancerous control [115,116,117]. When the purpose is to demonstrate that the toxic activity results from the association of the biogenic NP, it is imperative also to test the toxicity of the NP-free plant extract and determine how much of the effect is attributed solely to the extract [103,105,118].

The study of the cytoskeleton morphology after exposing cells to MNPs is important since the specific damage triggered to the cell structures can be known. In this sense, confocal laser scanning microscopy (CLSM) is a powerful technique that can be applied to analyze the effect of some NPs on cell morphology. This can provide information on the rearrangement of the cytoskeleton, which may be associated with the cell mobility and migration capacity, which is also related to metastasis. Proteins of the cytosketon play an important role in the curling or lamellipodia characteristics observed in cells during the migration or extension toward other surfaces [103,111]. For instance, green-based AuNPs inhibit the cytoskeletal rearrangement of the HT-1080 human fibrosarcoma cancer cell line [111].

Indeed, cell migration is a crucial step in cancer progression, particularly in the tumor growth and metastasis processes [119]. In cancer progression, invasion and metastasis occur when tumor cells disseminate from the primary tumor spreading through the circulatory and lymphatic systems, invade across the basement membranes and endothelial walls, and colonize distant organs [120]. Metastasis is a very harmful feature of some cancers. The migration assays are usually performed in transwell systems containing two areas in close contact, one upper and one lower, separated with a porous membrane. In this experiment, the cancer cells that can move are seeded in the upper side, and in the lower part, the chemoattractant is placed. The amount of cells that migrate from the upper compartment to the lower compartment is determined, either in the presence or absence of NPs [111]. Cell migration and invasion are significant components of metastasis. Indeed, it can be inferred that the cell migration assay is part of a successful antimetastatic therapy design [119,121]. In addition, the effect of MNPs on cell migration could be examined using the scratch wound healing assay [121].

The smart design of NPs involves a particular effect on a specific target when a physical or chemical stimulus is applied to a cell culture or organism. One of the most widely used stimuli is light, which, combined with NPs, triggers the so-called photothermal effect. In this approach, either cell cultures or laboratory animal tissues exposed to NPs (e.g., AgNPs or titanium dioxide) are treated with radiation, usually a 808 nm NIR laser [122]. In nanostructures such as gold nanorods (AuNR) [107], NIR radiation causes a local temperature increase that damages and kills cells. In addition to the MTT assay, this type of assay can also use fluorescent stains to demonstrate viability. The method uses Calcein for viable cells and propidium iodide that penetrates non-viable cells. This method is based on the calcein-acetoxymethyl ester (calcein-AM) [123].

Determining the internalization of MNPs is a fundamental requirement when a therapy design includes one or more intracellular targets. For this purpose, some methods allow an estimation of the amount of NPs that have been taken up. Inductively coupled plasma mass spectrometry (ICP-MS) measures nanograms per liter of metals or metalloids from biological samples after digestion in, e.g., aqua regia. Its high sensitivity has made it possible to identify cells with green biosynthesized MNPs [106]. Another assay often used to quantify the % of cells with internalized NPs is flow cytometry [107]. However, the nanosystem must involve a fluorescent tag conjugation or an antibody system to specifically detect the MNPs. It is also possible to determine MNPs’ internalization and location by CLSM, using a fluorescent marker that reveals the particle’s location. Liu et at. [105] reported a bacillary nanoparticle conjugated with a molecule capable of absorbing near-infrared radiation (indocyanine green dye, ICG) for microscopic confocal observation. This molecule emits infrared light (800 to 860 nm) when irradiated with a laser source between 740 and 800 nm. It can be used to localize nanoparticles in cells with adequate resolution. The combination of the Mito-Tracker Green and Hoechst 33258 (to contrast the cell nucleus) is used to confirm the colocalization of NP in the cell nucleus or organelles such as mitochondria [105].

One more perturbation that MNPs may cause to cells is oxidative stress, where several ROS participate. The evaluation of ROS levels is necessary to explain the damage observed after exposing cells to MNPs. For this purpose, precursors of fluorescent molecules such as 2′,7′-Dichlorodihydrofluorescein diacetate (DCFH-DA) are used. They are metabolized and subsequently oxidized by ROS, such as H_2_O_2_, and generate fluorescent molecules such as 2′,7′-dichlorofluorescein (DCF), the emission of which can be detected at 530 nm [124]. Green biosynthesized AgNPs from plant extracts or bacteria cultures have shown the ability to induce oxidative damage at the expense of ROS [102,106]. Another similar method uses the 1,3-diphenylisobenzofuran (DPBF) probe, which can emit fluorescence at 450 nm and reacts with OH radicals to generate the non-fluorescent derivate 1,2-dibenzoylbenzene (DBB). Thus, a decrease in the fluorescence signal determines the amount of ROS in a biological sample [125]. This proof has been successfully performed to demonstrate the ability of certain nanosystems to generate ROS in breast cancer cells after stimulation with near-infrared radiation (808 nm) [107].

Cancer cells are characterized by no longer being subjected to programmed cell death or apoptosis, a process that naturally limits the proliferation of cells. Finding alternatives that induce this process in cancer cells is one of the main objectives in developing anticancer treatments. Fluorescence-based assays are preliminary tests used to identify whether apoptosis is the phenomenon that destroys cells exposed to NPs [100,117]. Acridine orange and ethidium bromide are generally used for apoptotic testing. Cells allow the entry of ethidium bromide, which has a characteristic emission in the spectral range from 540 to 700 nm [126], appearing reddish-orange under the microscope. Viable, non-apoptotic cells show a fluorescent green color. Hoechst is a molecule that intercalates in double-stranded DNA in the AT-rich regions, emitting its typical blue fluorescence [127], which allows observation of the characteristic nuclear fragmentation of apoptosis [128]. Induction of apoptosis is determined mainly by two fluorescence-based tests: annexin V and caspase activation. Some commercial chemiluminescence kits have been developed. Apoptosis can also be quantified using flow cytometry. Some kits have been described using annexin-FITC and propidium iodide, based on detecting cells that externalize phosphatidylserine (PS). A critical step in this method is that the PS binding with annexin V is highly dependent on the Ca^2+^-concentration of the medium. Usually, necrotic cells with compromised membranes are counter-stained with propidium iodide exhibiting green and red fluorescence. Healthy cells are not stained. The evaluation is possible either by epifluorescence microscopy or by flow cytometry [129,130]. Thus, the presence of one or both fluorescence indicates either no viability, necrosis, or apoptosis, the latter in its early or late phase [130]. These experiments can be complementary. With these methods, it has been observed that when using AgNPs synthesized by the green pathway, the main damage is attributed to necrosis rather than apoptosis [106]. However, some other NPs clearly show that they can induce higher levels of apoptosis [105,107,131].

On the other hand, the relative expression profile of genes involved in apoptosis or during proliferative processes has also been evaluated to determine not only if NPs are capable to induce apoptosis, but also if they are able to enhance cell proliferation and/or the capacity for tumor progression. For this purpose, mRNA expression has been evaluated by RT-PCR, as well as the detection of proteins by Western blot. Some of the markers required to evaluate NPs include the following genes: fbw7a, p53, c-myc, skp2, bax, bcl2, and the proteins: Caspase 3, Caspase 9, Bcl-2, SRp30a c-myc, p53, Bax, Hsp70, and PARP-1 [118,131]. The cascades of caspases conduct the apoptotic process and can be divided into initiator, effector, and proinflammatory caspases. After cleavage of the peptide, the fluorochrome is separated from its quencher thus can be visualized in the cell by epifluorescence microscopy or even quantified by flow cytometry [132].

MNPs can also injure the mitochondrial membrane. This triggered damage can be monitored using the 5,5′, 6,6′-tetrachloro-1,1 ′, 3,3′-tetraethylbenzimidazolylcarbocyanine iodide (JC-1) dye. When cells are healthy, their mitochondrial membrane potential is high, but the potential decreases when there is any damage. The JC-1 is added to the cell culture, and, if the indicator emits reddish-orange fluorescence (590 nm), it means no damage to the mitochondrial membrane. On the other hand, when cells are damaged, the mitochondrial membrane will emit a green fluorescent light (515 nm ± 5 nm) [132]. The relationship between the green/red emissions obtained by flow cytometry indicates harmful effects at the mitochondrial level when cells are exposed to MNPs and/or photothermal treatments (taking advantage of the NPs SRP), and the effect caused by molecules sensitive to pH changes [107,131].

The cell cycle guides the processes of proliferation or quiescence in every kind of cell. Cancer cells present important alterations, for which it is assumed that substances that can affect the cell cycle could also have anticancer properties. Therefore, it is also crucial to consider evaluating the effect of NPs on the cell cycle. It has been shown that some biogenic AgNPs induce the arrest of the S1 cycle phase, which explains part of its anticancer effect [106]. Flow cytometry is used with markers for G0/G1, S, and G2/M phases [133].

Metabolic activity can also be a helpful marker in evaluating the harmful effect of MNPs on any cells. It could be beneficial if a treatment could decrease the high metabolic activity that cancer cells usually present. Commercial standardized tests have been used for this purpose and focus on quantifying ATP as a marker of metabolic activity, which also allows confirmation of the possible effects of some AuNPs with the ability to decrease the metabolism of cancer cells [131].

### 3.3. Mimetic 3D Cell Culture

Three-dimensional cell culture models have emerged to bridge the gap between 2D culture systems and animal models for the testing of new anticancer drugs [134]. These models closely resemble the in vivo TME and show structural and functional similarities to solid human tumors, such as the cell-to-cell interaction, a developed extracellular matrix (ECM), pH, oxygen, and metabolic and proliferative gradients [135]. Three-dimensional cell culture models are divided into scaffold-based, scaffold-free, and hybrids (Figure 4c). Scaffold-based models include hydrogel-based support, polymeric complex material-based support, hydrophilic glass fiber, and organoids. In contrast, scaffold-free models are produced through hanging drop microplates, magnetic levitation, and spheroid microplates with an ultra-low attachment coating. Hybrid culture systems combine microfluidic devices and micropatterned plates with ECM components with spheroids embedded inside ECM scaffolds [95,136,137].

Despite the numerous advantages offered by these models, they have been scarcely used to evaluate green biogenic MNPs. Recently, Vemuri et al. [138] reported the synthesis of AuNPs using naturally derived phytochemicals, such as curcumin, turmeric, quercetin, and paclitaxel, and their evaluation against two breast cancer cell lines (MCF-7 and MDA-MB 231). The synthesized NPs were found to be spherical and showed an average size ranging between 3 and 60 nm. These NPs effectively inhibited cell proliferation, angiogenesis, colony formation, and spheroid formation of breast cancer cells through apoptosis induction; HIF-1α, VEGF, Cyclin D1, and STAT-3 gene down-regulation; and Caspase-9 gene up-regulation. Ag-oxide NPs synthesized utilizing an aqueous leaf extract from *Excoecaria agallocha*, and with anticancer potential, were evaluated by Banerjee et al. [139]. These Ag-based NPs were spherical shaped and exhibited an average particle size of 228 nm. They induced a growth-inhibitory effect against murine Ehrlich ascites carcinoma cells, with an IC_50_ value of 1.1 ± 0.1 µg/mL at 72 h. Their cytotoxic effect was mediated via apoptosis induction as the number of annexin V-positive cells increased as a function of time [139].

For 3D models, spheroids’ integrity is very important. Therefore, after treatment, an enzymatic and mechanical spheroid dissociation is not recommended as it could alter the results and their interpretation. Several assays have been performed to assess the viability of spheroids treated with non-biogenic NPs, for example, fluorescent substances, such as Calcein-AM/ethidium homodimer-1 (EthD-1) (Live/Dead) combined with confocal laser scanning microscopy (CLSM) [140], CellTox^®^ Green dye combined with inverted fluorescence microscopy [141], fluorescein diacetate (FDA)/propidium iodide (PI) [142], or luminescent chemistry of CellTiter-Glo 3D^®^ [143]. Spheroid morphology is usually evaluated by optical microscopy, scanning electron microscopy (SEM), or confocal microscopy [140,144]. Moreover, several histochemical tests, such as hematoxylin/eosin, alkaline phosphatase, alizarin red, Masson’s trichrome, and aniline blue staining, have been applied to visualize internal biological structures [140,143]. In turn, many studies reporting gene expression determinations utilized semi-quantitative reverse transcription-polymerase chain reaction (RT-PCR) or quantitative PCR (qPCR) [142,144,145]. The evaluation of proteins expression is carried out by Western blot, immunocytochemistry, or bead-based multiplexed immunoassay tests [140,142,143,145,146]. These, and other methods, can be prospectively used to characterize in more depth the effects of green biosynthesized MNPs on tumor spheroids.

### 3.4. Ex Vivo Models

The poor prognosis of some types of cancer is attributed to the complex tumor-tissue multicellular microenvironment, as it is difficult to completely mimic it in preclinical models. Therefore, developing and maintaining ex vivo models of cancerous tissues that preserve the structure, multicellular 3D architecture, and viability of each type of human cancer signaling remains challenging. Understanding this type of explants and the impact that external stimuli may have on tumoral tissues could accelerate complementary therapies and favor personalized medicine [147]. Another approach that provides a better in vivo-like environment is the precise cutting of tissue sections, representing an ex vivo model of the study organ while maintaining the original architecture. The advantage of this system is that sections from different species, such as rodents and human biopsy material, can be prepared and compared. Most of these studies have focused on pharma-toxicological studies [148].

Despite the friendly properties of biosynthesized MNPs, it is also essential to validate their safety and biological effect in complementary experimental models, as has been carried out for non-biogenic NMs. Moreover, ex vivo models are necessary to confirm the essential characteristics of the MNPs. Kokkinos et al. [147] developed a new preclinical model of pancreatic ductal adenocarcinoma that preserves for 12 days the native 3D multicellular architecture of human pancreatic tumors in culture. Furthermore, they demonstrated that this tissue explant model is susceptible to transfection with gene silencing based on polymeric NPs, delivering siRNA to pancreatic adenocarcinoma cells in vitro and in vivo. Gokulan et al. [149] reported the performance of an ex vivo model using human intestinal tissue to evaluate changes in levels of pro-/anti-inflammatory cytokines/chemokines and mRNA expression of intestinal permeability-related genes induced by commercially available AgNPs in ileal tissues.

The research group headed by Gonzalez aims to evaluate and elucidate the signaling pathways that various NMs confer, delineating a physiological profile using ex vivo models. According to their experience and other research, using ex vivo physiological models in the study of AgNPs and AuNPs can exert variations in the physiology in similar patterns to endogenous hormones or mediators [150]. The ex vivo physiological models of isolated tissues and organs that are used to evaluate NMs allow the evaluation of particular functions. Some examples are ducts related to the cardiovascular, respiratory, and digestive systems (small or large intestines) or the study of organs, such as heart, kidney, lung, liver, and the biochemical communications involved between organs and tissues [151,152]. The results obtained from these investigations have allowed us to elucidate the mechanisms of action triggered by this type of NM in various biological structures, reorienting future research to gain knowledge concerning their beneficial or toxicological effects and establishing the toxic values (maximum permissible dose, MPD) for regulated biomedical applications.

#### 3.4.1. Isolated Tissue System

The isolated tissue system represents a classic method for investigating the physiology and pharmacology of isolated blood vessels, airways, and intestines by applying isotonic or isometric measurements using appropriate transducers. The isometric measurement is used to assess contraction by keeping tissue length constant, whereas, in the isotonic mode, the dimension of the tissue is decreased by an applied force. For instance, coupled to isometric transducers, the isolated ring system is used primarily to monitor tension in small tissue sections and rings in real-time [153,154]. Additionally, in the physiological solution that contains the ring, under a given treatment, the presence of various mediators or molecules produced from that treatment can be determined and quantified [155,156,157]. Indeed, murine models allow the study of the structure and function of various organs, both under physiological or pathological conditions, and constitute an essential tool in analyzing NM-induced responses. The experimental models exposed involve rodents, such as mice, rats, guinea pigs, or rabbits.

#### 3.4.2. Isolated and Perfused Organ System

Isolated organs, such as the heart, kidney, and liver separated from an in vivo system, e.g., experimental rodents, can maintain their viability, functionality, and metabolic processes for a limited time. In this period, the organ under study sustains physiological and biochemical parameters that provide basic knowledge about the behavior of a substance at the organic level without having the interference of other structures and mediators from an in vivo system. The isolated and perfused organ maintains its functionality due to the physiological solution that passes through the blood vessels and irrigates the organ under study, keeping the physiological conditions of oxygenation, pH, and temperature [158]. Langendorff’s isolated and perfused heart model has lead to fundamental insights into cardiovascular biology and physiology [159]. The basis of Langendorff’s isolated heart model is to maintain cardiac activity by perfusing the heart through the coronary arteries using an aortic cannula inserted into the ascending aorta. Thus, the infusion solution enters retrogradely into the heart through the aortic cannula. The retrograde perfusion is produced by hydrostatic pressure (constant pressure model) or by using a pump (continuous flow model) that closes the aortic valve; in this way, the perfusion solution flows through the aorta contrary to in vivo conditions [160,161].

#### 3.4.3. Integrative Evidence about the Physiological Profile of AgNPs and AuNPs

Information regarding AgNPs, such as their biophysical properties, functions, effects at different levels of biological organization, and their impact on human health, is still controversial. In recent years, our laboratory has investigated the biological effects triggered by AgNPs at different biological levels and their possible toxic or beneficial implications in the cardiovascular and respiratory systems [162,163]. Gonzalez et al. [164] observed, in an isolated ring system of the aorta, that AgNPs exhibit a series of events as a function of concentration, shape, and size. For example, spherical 37.5 nm AgNPs induce two types of effects: at low concentrations, vasoconstriction is increased in the isolated rings precontracted or not, while at higher concentrations, a vasodilator effect was observed in phenylephrine precontracted aortic rings. Therefore, it was suggested that AgNPs might block the action of powerful vasodilator agents such as the acetylcholine (ACh) produced in the body. At higher concentrations, AgNPs stimulate vasodilation mediated by the activation of endothelial nitric oxide synthase (eNOS), which produces low concentrations of nitric oxide (NO), an important vasodilator and antihypertensive agent [164]. Likewise, this effect depended on the endothelium (E) (the inner layer surrounding the blood vessels). NO production was suppressed when this was removed, and the vasodilator effect was lost [165]. A similar effect occurred in isolated and perfused hearts, similar to those observed in blood vessels. With these physiological effects, it was possible to evaluate the responses induced by AgNPs and to study the possible mechanisms of action involved.

The physiological profile of AuNPs is also interesting, as it focuses on biosafety, biocompatibility, and the design of drug delivery systems. Some evidence displays the effect of AuNPs in ex vivo models: i) Silva et al. [156] reported how AuNPs modify NO release and vasodilation in rat aorta. The authors investigated the role of ruthenium (NO donor), AuNPs, and an AuNPs–ruthenium composite in isolated rat aortic rings, demonstrating that it is possible to modify the NO release profile. All these systems favor relaxation of the aorta, although each system selectively and explicitly activated processes related to the metabolism of NO and potassium (K) channels. The results provided insights into the role of AuNPs and their functionalization as a pharmacological strategy to control NO levels. To improve cancer treatment, one alternative could be to locally increase the NO concentration that should promote cytotoxic effects and vasodilation, thus contributing to the involution of tumors. In addition, administration at low concentrations to increase the normal concentration of NO through the tumor’s blood supply, which may impair the dilation of the capillaries and restrict blood flow, could be a promising strategy for tumor growth inhibition. ii) Mohamed et al. [155] reported AuNPs synthesized by the Turquevich method without a significant effect on ACh vasodilation. Nonetheless, they can block the relaxation induced by sodium nitroprusside (SNP), but the PVP-modified AuNPs attenuated ACh-induced dilation. The incubation with PVP alone promoted a significant reduction in ACh responses. In comparison, vessel incubation with the PVP-modified AuNPs induced a significant decrease in SNP responses. When vessels were incubated in PVP alone, a non-significant effect on SNP responses was noticed, suggesting that the AuNPs per se may be interfering with the action of SNP. iii) Maldonado-Ortega et al. [150] showed that AuNPs at 100 µg/mL induce a contractile action on isolated rings of rats’ trachea, where NO is a potential mediator. This work contributed to a better understanding of the participation and association of NO in contractile processes and tracheal hyperresponsiveness. In conclusion, even though there is no evidence on the ex vivo study of NPs synthesized by green biological methods, this type of models can provide important information for their biological and physiological evaluation and validation. They offer mensurable knowledge about the possible mechanisms of action for preclinical evaluation and application. In the biomedical field, the impact of the ex vivo approaches could substantially impact the development and design of new pharmacological strategies against cancer.

### 3.5. In Vivo Evaluation Models

Several studies aim at mimicking the characteristics of the tumor using in vivo models. Their significance for cancer research lies in the possibility of knowing the biology of cancer to develop new therapies. Different animal models have been established as notable tools to study human cancers, providing valuable information on the biology of cancer, the evaluation of new antitumor therapies, the discovery of target molecules, and the validation of biomarkers [166]. Current research continues to look at a broad spectrum of cancers with the aim of understanding their biological behavior. In this sense, animal models must include relevant characteristics of the tumor, such as its microenvironment, anatomy, natural history, angiogenesis, and metastasis. Furthermore, animal models are crucial to understand the pharmacokinetics, metabolism, and distribution of antineoplastic drugs [166,167]. Technological advances in genetic and cancer tissue engineering offer enormous potential information from preclinical models [168]. Therefore, summarizing the most recent advances in in vivo models using MNPs offers a value platform of preclinical nanomedicine evaluation. This review focuses on the soil nematode *Caenorhabditis elegans* (*C. elegans*), the freshwater fish *Danio rerio* (*D. rerio*) known as Zebrafish, and the murine model.

#### 3.5.1. *C. elegans* and *D. rerio* (Zebrafish) Models

*C. elegans* and zebrafish models have been used to understand fundamental biological processes involved in cancer, such as apoptosis, proliferation, angiogenesis, invasion, metastasis, genome instability, and metabolism. *C. elegans* offers a powerful platform for studying carcinogenesis and identifying new cancer drug targets [168]. *C. elegans* shares a high homology with human genes. Many biological processes, including apoptosis, cell signaling, cell cycle, cell polarity, metabolism, and aging are conserved between *C. elegans* and mammals [169]. Zebrafish is a valuable model widely used to study developmental biology and cancer. The evolutionary conservation of cancer-related programs between humans and zebrafish is surprising and allows the results obtained in fish to be extrapolated to humans. Zebrafish is a reliable model to study human cancer, as recent xenotransplantation studies in zebrafish have shown to be adequate for evaluating the invasiveness of patient-derived xenograft cells [170]. These organisms are excellent models due to the great variety of genetic, molecular, and biochemical tools available for their study and the significant conservation of their genes [168,170]. Table 2 summarizes some research works related to this topic.

For instance, studies exposing larval-stage nematodes to AuNPs revealed several differentially expressed genes. The majority were up-regulated and related to the amyloid processing, citrate cycle, clathrin-mediated endocytosis, apoptosis, and G-protein signaling. These findings suggested that the *C. elegans* AuNPs uptake is achieved by endocytosis via clathrin coating. AuNP exposure also induced neural damage and changes in the feeding behavior. Furthermore, mutant animals showed hyper sensibility to AuNPs [171]. Other studies [167,172] showed that AuNPs triggered changes in the cellular defense response and lipid catabolic processes of *C. elegans*. Additionally, changes in lipid storage, body morphogenesis, shape, and size were observed. The processes of detoxification of metals, homeostasis, and adaptation to stress were also modified. They also showed morphological changes in the offspring, locomotion problems, and fertility alterations.

On the other hand, when adult zebrafish were exposed to AuNPs for 96 h, the gene expression at the lowest concentration was similar to that of the control. The authors found that down-regulation affects biological processes related to development, biogenesis, metabolic processes, cellular localization, biological adhesion, and locomotion [173]. A different study demonstrated that when zebrafish larvae are chronically exposed to AgNPs until the stage of adulthood, the fish present affections in the locomotion, fertility, cell growth, and neuroactive interaction, showing lower locomotion and small body lengths. Chronic exposure (28 days) of adult zebrafish to AgNPs showed changes in the extracellular components related to the extracellular matrix, mitochondria, and ribonucleoprotein complexes, also causing damage to DNA. Unexpectedly, the zebrafish gills of treated animals did not show morphological defects despite changes in the expression of several genes [175]. Similarly, zebrafish exposure to AuNPs showed effects on locomotion velocity, growth, and reproduction [174].

Studies performed in zebrafish larvae, in which the exposure to AgNPs was carried out during six days post-fertilization, showed no adverse effects on fish survival and growth. Unexpectedly, AgNP exposure resulted in higher survival rates for zebrafish larvae, particularly with the highest concentration (1 mg/L) [176]. Other studies identified a strong accumulation of Ag in the blood vessels of the liver, in the interstitial tissue [178], and neural changes after AgNPs exposure [173]. The overlapping functions were altered when nematodes or zebrafish were exposed to MNPs, among these: cell signaling (MAPK signal or G protein), control of cell growth, apoptosis, stress response, and DNA damage [171,172,173,174,175,176]. Most of these responses can trigger cancer, demonstrating that these model organisms are very useful to study the impact of NPs at the level of the whole organism. Interestingly, MNPs significantly impact the expression of genes related to development and neurogenesis. Altering the expression of developmental genes could lead to misregulation of pathways that can cause a malignant formation. *C. elegans* and zebrafish could be used for future approaches or as a preclinical cancer model alongside mouse use.

#### 3.5.2. Murine Model

The murine model will be briefly described because it is an excellent organism to study cancer onset, invasion, and metastasis. It represents a significant step between in vitro systems and clinical studies [179]. The mouse genome is highly homologous to the human genome, which can simulate a series of biological characteristics, such as the occurrence, development, and metastasis of human cancer cells in vivo [180]. Moreover, it has the advantages of convenient feeding, low price, and easy gene modification. It provides a good tool for cancer research and drug discovery and verification [181]. The most widely accepted animal models in cancer research are syngeneic, genetically modified mouse models (GEMMs), chemically induced models, and xenograft models. Xenografts can be divided based on the source of the tumor: xenografts with conventional cell lines (cell line-derived xenografts, CDX) or with the use of samples obtained from patients with some kind of cancer (patient-derived xenografts, PDX). Table 3 describes the advantages and disadvantages of the main in vivo murine models for cancer research.

In GEMMs, spontaneous tumor initiation occurs within the correct microenvironment from an otherwise normal tissue cell. These may be simple oncogenic-driven transgenic mice. One limitation to the conventional GEMM models is that the regulatory sequences used to drive transgene expression are not well defined in specific lineage/expression domains. The specific oncogenes may not necessarily reflect those observed in human tumors. Nonetheless, these models serve a purpose in cancer research [182]. This field has turned to more specific models emulating the genetics of human disease with spatial and temporal activation of oncogenes and deletion of tumors suppressors targeting mouse tissues [183].

A cell line-derived Xenograft or CDX model is widely used to test anticancer therapies. Human tumor samples are cultured as cell lines and implanted into immunodeficient nude animals to test the efficacy of antitumor compounds in vivo. CDX is one of the simplest, easiest, and most commonly used systems based on the engraftment of human cancer cell lines to immunodeficient animals [184]. CDX has proven to be very useful for probing cancer genetics, biological processes, and metastatic potential. However, it has some limitations that include reduced intra-tumoral heterogeneity and low effectiveness in predicting clinical performances. In addition, the lines used are frequently derived from highly aggressive malignant tumors, making these less useful for modeling early events in the evolution of the primary tumor. Moreover, in most of the cases, it is necessary to use immunosuppressed animals, increasing the cost of the animals’ care. The transplant site is also a critical issue to consider. Generally, subcutaneous injection (ectopic) and the implantation of cells in the specific tissue of the mouse (orthotopic) are used [184].

It is worth mentioning the wide acceptance of the PDX models in the pharmaceutical industry. Indeed, transplanting tumor pieces or primary human cancer cells into host mice is of clinical relevance. The transplantation of human tumor fragments into immunocompromised mice has been reported by Hoffman et al. [184]. PDXs are very attractive models due to the preservation of many relevant characteristics of the primary human tumor, such as growth kinetics, histological characteristics, and behavioral characteristics (invasiveness and metastatic capacity), regardless of the ability to respond to tumor therapy [185,186,187,188,189,190]. Moreover, these models have had an important industrial impact, and they are the choice for translational research.

On the other hand, many studies show that AuNPs and AgNPs obtained by green biosynthesis have cytotoxic or antiproliferative effects on different tumor cells of different types of cancer. However, most of these studies are carried out with cells grown in vitro. The evaluations of antitumor activity using in vivo models are relatively scarce. It is relevant and essential to carry out in vivo evaluations of the potential use of AuNPs and AgNPs as soon as possible since these models are closer to cancer in patients. Although there have been advances in establishing diagnostic and therapeutic applications of AuNPs and AgNPs synthesized by chemical methods, it is also necessary to evaluate biogenic nanoparticles. Table 4 summarizes some studies performed to evaluate the antitumor properties of Au and AgNPs employing murine in vivo models.

The design and selection of the exploratory or confirmatory evaluation models are still challenging, and the influence of the immune system, genetic factors, lifestyle, and environmental factors must be considered. Figure 5 showed the main goals for using in silico, in vitro, ex vivo, and in vivo models for MNP evaluation.

## 4. Application of Biosynthetic MNPs in Cancer Therapy

Until now, numerous papers have been published regarding the use of non-biogenic Au and AgNPs in cancer applications using in vitro, ex vivo, or in vivo models. The green biosynthesis of AuNPs and AgNPs is an eco-friendly and low-priced process that provides biocompatible nanoagents with potential anticancer activities. This section aims to provide an overview of the advances in evaluating the antitumor effect of biosynthesized MNPs, particularly Au and AgNPs, against different types of cancer: skin, breast, lung, prostate, colorectal, cervical, and leukemia.

### 4.1. Skin Cancer

The skin is the largest organ of a mammal’s body. Its exposure to the environment makes it highly susceptible to physical, chemical, and biological perturbations that frequently trigger cancer development [193]. Skin cancer is the most common type of any form of cancer. It is classified by the origin of the initial cells, where non-melanoma skin cancer is the most common form (70% of the total cases), and the melanoma type occurs in smaller proportion (30% of the cases and represent the fourth form of the worldwide new cases of cancer) [194]. In the last year, new cases of non-melanoma skin cancer worldwide affected 1.2 million people, with 2.5% deaths. Melanoma begins as a focalized and limited lesion [195,196]. However, together with the Merkel cell carcinoma, melanoma could be metastatic [197,198]. The primary skin cancer treatment is surgery, with or without radiation. When the surgical procedure is not enough to remove the whole tumor, or when malignant cells spread out from the dissection zone, chemotherapy is also necessary to limit or eliminate the tumor [199]. It is necessary to improve drug administration, either for single chemotherapy or photodynamic therapy (PDT), where the lesion is permeated with a photosensitizer and then exposed to visible light [200]. After conventional cancer treatment, and to prevent new lesions in the susceptible skin or beyond, different substances must be deposited below the epidermal dead cell layers [201].

Many MNPs have been designed and produced with the aim of skin cancer treatment by binding different drugs or extracts to their surface. For instance, an aqueous extract from the roots of *Siberian ginseng*(SG) showed anticancer properties against murine melanoma B16 cells. It induced apoptosis through Bid, Bad, Casp-3, and Casp-9 gene overexpression [202]. AgNPs synthesized using *Anona muricata* and with an average diameter smaller than 50 nm showed interesting properties in that they destroyed melanoma cancer cells A-375 at concentrations lower than 3 µg/mL [203]. Safwat et al. [204] prepared AuNPs loaded with 5-FU as a transdermal delivery system. This cream or gel-based formulation (5-FU/CTAB-AuNP) was assessed in a mouse skin cancer model, showing a diminished tumor weight compared to controls. Its effectiveness was vehicle dependent, where cream significantly reduced the pathological alterations and a near-complete regression of epidermal and dermal infiltrations compared to the gel form.

In addition, PDT has been proposed as an effective modality for melanoma therapy because of its target-specific effects. For example, titanium dioxide nanoparticles (TiO_2_ NPs), gold nano-clusters, and graphene can elicit a series of toxicological responses in mouse B16F1 melanoma cells when stimulated in sunlight. This is due to the depolarization of the mitochondrial membrane and the generation of superoxide radicals [205]. Furthermore, AgNPs coupled with 5-aminolevulinic acid (5-ALA) exhibit cytotoxicity in skin melanoma B16F10 cells and squamous cell carcinoma A431 cells when are briefly exposed to halogen light. In this case, cell toxicity responds to tumor aggressiveness and ROS production [206]. On the other hand, photothermal therapy kills cancer cells by using the heat generated from absorbed near-infrared energy with minimum damage between the tumor and the surrounding tissues [207]. AgNPs coated with TiO_2_, produced by a sol-gel two-step technique, demonstrated their capability of eliminating subcutaneous melanoma tumor in vitro and in vivo in B16-F10 cells and C57BL/6J mice, respectively, when exposed to near-infrared radiation [122]. Recently, bovine serum albumin (BSA)-coated silver NPs (BSA-AgNPs) have been developed to generate free radicals that were likely derived by oxidative stress to B16F10 cells [206]. In a recent work, Liu et al. [206] developed a light-inducible nucleic acid delivery Ag-based nanosystem. These AgNPs released miR-148b that induced apoptosis in Ras expressing keratinocytes and murine squamous cell carcinoma cells while avoiding cytotoxicity in untransformed keratinocytes. In summary, AgNPs and AuNPs have been efficiently synthesized following green approaches and evaluated for their potential application for skin cancer therapy, where the size, shape, and evaluation models have been validated through different assays (see Table 5).

### 4.2. Breast Cancer

Breast cancer is the first in frequency among all types of cancer. Of all breast cancer cases, 99% occur in women (1% in men). The incidence of breast cancer cases and deaths has been increasing in the last 20 years. It is significant and relevant, from 1.15 and 0.410 million in 2002 to 2.26 and 0.685 million in 2020, respectively [195,210]. A variety of breast cancer treatments exist and are available at every development stage. Most patients require a combination of two or more treatments. After diagnosis, doctors determine the stage of cancer. They then decide on the best treatment options based on the stage and other factors, such as age, family history, genetic mutation status, and personal medical history. Breast cancer can be treated with surgery (mastectomy or breast-conserving surgery, i.e., lumpectomy, quadrantectomy, partial mastectomy, or segmental mastectomy), radiation therapy (high-energy X-rays), chemotherapy (docetaxel, doxorubicin, and cyclophosphamide), hormone therapy (hormone-receptor-positive), and targeted therapy [211,212,213]. In this sense, monoclonal antibodies (e.g., trastuzumab-Herceptin), antibody-drug conjugates (e.g., ado-trastuzumab emtansine, Kadcyla, or TDM-1), and kinase inhibitors (e.g., lapatinib and other inhibitors) are often used in targeted therapy [214]. The management of breast cancer in older women is highly individualized and requires collaboration across disciplines (medical oncology, surgical oncology, and radiation oncology) [212]. In the last 20 years, treatments have been employed as a key tool in controlling breast cancer, with a 65–80% sensitivity and specificity [213]. Therapies are not sufficiently developed as they have certain limitations. The most common is their non-specificity between normal and cancer cells, resulting in inevitable side effects and poor effects at the III and IV cancer stages [211]. Treatments for early-stage breast cancer may not be effective for advanced-stage breast cancer. If breast cancer is detected and the therapy applied when the tumor is confined to the breast, remission can reach nearly 100%. Unfortunately, small breast tumors are rarely detected by a physical examination in the early stages. Sometimes, they may not be observed in a mammography analysis, particularly in young women and women with dense breast tissue [212,215,216].

Au and AgNPs have shown a very interesting potential for breast cancer therapy and drug delivery in both in vitro and in vivo systems. Several MNPs obtained by green bioprocesses have been evaluated for potential application against breast cancer. The MCF7 and MDA-MD-231 in vitro models are commonly used for breast cancer cytotoxicity evaluation, with the MTT assay being the preferential cytotoxicity and viability test (see Table 6). Plant extracts are mainly used to obtain Au and AgNPs by the biological green route for breast antitumor activities. Most of the MNPs obtained are spherical with diameters below 100 nm. The cytotoxicity displayed by these NPs is significantly higher for breast cancer cells than healthy cells. However, the molecular mechanisms involved in the biogenic Au and AgNP-induced cytotoxicity against breast cancer cells are not fully understood; some studies proposed ROS generation and apoptosis [217]. Although the data obtained from in vitro models are promising, further in vivo investigations are required to demonstrate the reliability and efficacy of these NPs in animal models.

### 4.3. Lung Cancer

Lung cancer is the most commonly diagnosed cancer. The incidence of lung cancer cases and deaths has been increasing significantly in the last 20 years, from 1.35 and 1.179 million in 2002 to 2.2 and 1.8 million in 2020, respectively [239]. Tobacco smoking is recognized as the major cause of lung cancer. Other known risk factors include idiopathic pulmonary fibrosis, chronic obstructive pulmonary disease (COPD), personal or family history of lung cancer, and the exposure to several occupational and environmental carcinogens, such as arsenic, radon, asbestos, and polycyclic aromatic hydrocarbons (PAHs) [240]. Current lung cancer treatment modalities include surgical resection, chemotherapy, radiation, and immunotherapy [241]. Several Ag and AuNPs have been synthesized via green biological approaches, seeking a potential application against lung cancer (Table 7). Aqueous leaf extracts have been mostly used for NPs synthesis. However, other plant materials, such as bark, pericarp, stem, needle, root, peel, flower, seed, and gum have also been used. Moreover, microalgae and bacteria have been useful for this purpose. Formed NPs are mostly spherical-shaped and exhibit cytotoxic effects with IC_50_ values <100 µg/mL. Interestingly, some of them were cytotoxic at low micromolar concentrations. The most commonly used in vitro model is the human lung epithelial carcinoma cell line A549. Many reports suggest that biosynthesized NPs exert their cytotoxic effects on lung cancer cells through apoptosis induction. For example, treatment of A549 cells with AuNPs synthesized using a *Rabdosia rubescens* aqueous leaf extract increased the proapoptotic Beclin-1, Bid, Bax, and caspase expression level 3 proteins, as well as caspase 3 and caspase 9 activity, whereas it decreased the expression of Bcl-2 protein. Moreover, DNA laddering was also observed [242]. On the other hand, AgNPs synthesized from *Pinus roxburghii* needle butanol fraction increased ROS levels, mitochondrial depolarization, nuclear condensation, DNA fragmentation, caspase 3 activation, and PARP-1 cleavage on A549 cells as reported by Kumari et al. [243]. Similarly, AgNPs synthesized from *Pleuropterus multiflorus* aqueous root extract induced DNA damage and activated caspase 3, p53, p38, and ERK expression on lung cancer cells [244].

### 4.4. Prostate Cancer

Prostate cancer (PCa) is a gender-specific global disease. It is the most frequently diagnosed type of cancer and the second cause of cancer deaths (7.1%) among males. According to the GLOBOCAN statistics, around 1.414 million of new cases and 0.375 million of deaths were attributed to PCa in 2020 [195]. PCa mortality is related to many factors, including poor early diagnosis, resistance to the treatment, and development of androgen receptor mutations [264], which is the main molecule used in the fight against PCa. The most widely used approach to treat PCa is based on reducing the level of androgens. It promotes the exacerbated growth of prostate cancer cells when it is over-activated. This therapy involves i) either the partial or total removal of the prostate [265]; ii) chemotherapy including the use of hormonal or non-hormonal drugs [266] for the androgen deprivation therapy (ADT), for example, docetaxel (Taxotere), cabazitaxel (Jevtana), and mitoxantrone (Novantrone) [267]; or iii) immunotherapy with Sipuleucel-T, a new dendritic cell vaccine designed to enhance the immune system of PCa patients [268,269]. All of these modalities can be used alone or in combinations and their success depend on factors, such as age, stage, and cancer progression. Cancer cells usually do not respond to ADT, becoming resistant and growing even without androgen stimuli.

At present, there is no effective therapy for PCa. Nanotechnology arises as an alternative to develop and use NMs as carriers or therapeutic agents with enhanced properties for a safety and effective treatment [270]. There are NMs composed of gold, silver, iron, zinc, or titanium containing agents with diverse compositions and therapeutic capacities e.g., natural compounds from plants [261], hydrolyzed peptides from proteins [262], and bacteria supernatants [263]. In recent years, these NMs have been improved by the use of green synthesis technology [259,260]. Green biogenic Au and AgNPs are highly versatile agents against PCa. These NPs present cytotoxic effects against human prostatic carcinoma cells, such as PC-3, LNCaP, and DU-145, cell lines used as PCa in vitro models. Death mechanisms triggered by NPs mainly depend on their bioactivity. In some cases, cellular responses improve ROS generation and promote cell death through apoptosis, activating caspase-3 and PARP-1 [243].

El Raey et al. [271] reported that AgNPs, synthesized using an extract from *Acalypha Wilkesiana* flowers, are cytotoxic to PC-3 cells. This study suggested that the presence of biapigenin derivatives, such as amentoflavone and cupressuflavone, which display a unique binding docking score towards the active site of the human DNA topoisomerase enzyme, cause DNA damage and cell death via apoptosis. Moreover, AgNPs synthesized using the *Dimocarpus Longan Lou* peel aqueous extract showed cytotoxicity against PC-3 through apoptosis by decreasing stat 3, bcl-2, and survival and by increasing caspase-3. Table 8 summarizes some Au and AgNPs generated by the biological green route and the cytotoxic effect against PC-3, LNCaP, and DU-145 prostate cancer cells.

### 4.5. Colorectal Cancer

Among all types of cancer, colorectal cancer is the third most common worldwide. It is found at around 49% in women and 51% in men. The incidence of colorectal cancer cases and deaths has been increasing in the last 20 years, from 1.02 and 0.529 million in 2002 to 1.89 and 0.91 million in 2020, respectively [195,210]. Colorectal cancer treatment depends on several factors. These include the size and location of tumors, cancer stage, whether the cancer is recurrent, and the patient’s overall health. Treatment options include i) chemotherapy, destroying cancer cells throughout the body using 5-FU, Capecitabine (Xeloda), Irinotecan (Camptosar), Oxaliplatin (Eloxatin), Trifluridine, or Tipiracil (Lonsurf); ii) targeted therapy using Bevacizumab (Avastin), Cetuximab (Erbitux), or Panitumumab (Vectibix); iii) immunotherapy through Pembrolizumab (Keytruda), Nivolumab (Opdivo); iv) radiation therapy; and v) surgery [293]. In general, chemotherapy and radiation therapy are more severe than targeted therapy, which enhances the selectivity of the treatment by targeting specific cells. Immunotherapy helps the body to use its immune system to detect and eliminate cancerous cells; people with advanced colorectal cancer are suitable candidates for this adjuvant therapy. Radiation therapy uses high-energy radiation beams to destroy cancer cells and prevent them from multiplying. It can have long- and short-term adverse effects. Ablation approaches involve the use of light, microwaves, radiofrequency, or cryosurgery to destroy a tumor without removing it [294].

The use of MNPs obtained by green biosynthesis may be very useful for solving the limitations of conventional therapies. Several Ag and AuNPs have been biosynthesized and evaluated for their potential application against colorectal cancer (Table 9). The HT29 and HTC116 cell lines are the most commonly used in vitro models for NP toxicity evaluation. Au and AgNPs have shown tumor-suppressive effects followed by DNA damage, mitochondrial dysfunction, cell-cycle arrest, and aberrant regulation of p53 effector proteins, which induce apoptosis in a dose-dependent manner [102]. Aqueous plant extracts, algae, and bacterial lysates have been used to synthesize Au and AgNPs (see Table 9 examples). The obtained NPs have sizes ranging from 1 to 100 nm, with an average size of 34 nm. MTT is the most widely employed assay to evaluate the toxicity induced by Au and AgNPs and thus viability of colorectal cancer cells [295,296]. ROS determination, protein expression profile, caspase activity, DNA analysis [297,298,299], and Annexin V [300] are other common assays.

### 4.6. Cervical Cancer

In 2002, 493,243 cases and 273,505 deaths caused by cervical cancer were reported. Worldwide, 604,000 new cases were detected and ~341,800 deaths of cervical cancer were recorded in 2020. Therefore, after breast, lung, and colorectal cancer, it is the fourth most common cause of death by cancer in women (7.7%) [195]. Sexually transmitted infection with Human Papilloma Virus (HPV) is a necessary but not sufficient cause for the development of cervical cancer. The variants HPV16 and HPV18 count globally for ~70% of all novel cases. Other risk factors include reproduction, long-term use of contraceptives, smoking, and obesity [311]. The standard treatments to cure cervical cancer are surgery and radio- and chemotherapy; the latter are frequently applied post-operatively as adjuvant therapies. Nevertheless, as those therapies are not specific to malignant cells, they affect healthy cells and cause adverse side effects.

Nowadays, this is the field where nanomedicine may contribute to the design of more effective protocols [311]. Until now, apart from composite nanovesicles, MNPs such as Ag and AuNPs have mainly been studied. Additionally, by being small enough (1–100 nm) to enter cells or cross biological barriers, NPs have high surface areas (~60 m^2^/cm^3^) that can be activated with specific components to selectively reach tumor cells [311]. In many human cancer cells, the folate receptor is overexpressed. Hence, much effort has been spent on developing nanomedicines using folate as a ligand [312]. A globally spread drug delivery system consists of using composite nanovesicles or nanopolymers that encapsulate approved cytotoxic agents such as cisplatin, carboplatin, paclitaxel, methotrexate, or topotecan and are decorated with folate for an active drug release. Their usefulness has been proven in several in vitro and in vivo studies. For instance, HeLa (cervix adenocarcinoma cells) internalized ~35% more nanopolymers functionalized with folate compared to those without the ligand [313]. This explains why nanopolymers containing either CBP (carboplatin)/PTX (paclitaxel) or DOX (doxorubicin) and folate as ligand significantly reduced the viability of HeLa cells by 70–77% compared to those without folate, which reduced their viability in about 53–55%. However, even without folate, nanopolymers are more effective than the corresponding free drug that reduced HeLa viability by 30–31% [313,314]. Similar results were observed in vivo using a mouse xenograft model with a HeLa-induced tumor. After 24 h, only ~3 µg/g tissue of the free drug reached the tumor. The delivery increased to ~8–9 and 10.5–12.5 µg/g tissue by using the bare and functionalized nanopolymers, respectively [313]. Moreover, while free cytotoxic agents cause adverse effects to other (healthy) internal organs, these side effects were significantly reduced when the cytotoxic agents were carried by the nanopolymers [313,314]. Nevertheless, in many chemotherapies, only a single therapeutic agent is used to eradicate cancer cells. The high tissue and genetic heterogeneity of cancer patients make this single-drug strategy ineffective. Hence, to enhance the anticancer treatment’s efficacy, different therapeutic agents may be combined. Pre-clinical and clinical studies still require more research to optimize this approach [315]. A promising alternative is the use of green biosynthesized NPs, e.g., nanoparticles coated with herbal extracts. Plant extracts are preferred over bacterial metabolites, which require additional purification to avoid bacterial contamination or proinflammatory reactions due to residual components. Gold and silver compounds are widely used to synthesize NPs because they are chemically inert and nontoxic [316]. Moreover, they have advantageous biological properties such as antibacterial, anti-inflammatory, and antioxidative effects [317]. Until now, the use of green biogenic Au and AgNPs against cervical uterine cancer has only been evaluated in vitro using mainly HeLa cells. These reports are summarized in Table 10. The cytotoxic, or at least the cytostatic, effect of NPs is typically measured in vitro in a range from 1 to 100 µg/mL. A proposed NPs cytotoxic mechanism consist of necrosis or apoptosis induced by ROS, and the activation of caspase cascades. Surprisingly, many studies lack an important control: the effect of the respective plant extract in the absence of NPs to determine a synergistic effect of green biosynthesized NPs vs. plant extract or naked nanoparticles.

### 4.7. Leukemia

Leukemia includes cancers characterized by disorders of malignant bone marrow cells that promote replacement with immature and undifferentiated abnormal hematopoietic cells so that they do not remove necessary cells, such as red blood cells, healthy platelets, and mature white blood cells (leukocytes), etc. [332]. Four main subtypes of leukemia are known: chronic lymphoid leukemia (CLL), chronic myeloid leukemia (CML), acute lymphoid leukemia (ALL), and acute myeloid leukemia (AML). AML and CLL are rare in children, while ALL is predominant in them. AML is common in people of all ages but is more common in adults. These subtypes represent different diseases, which vary in prognostic etiology, frequency of genetic abnormalities, and tolerance to chemotherapy [333]. In 2002, the number of cases was 300,522, and there were 222,142 of deaths [208]. Globally, in 2020, leukemia was classified as the fifteenth most frequently diagnosed cancer, with 474,519 cases and 311,594 deaths, constituting the eleventh cause of death from malignant disorders [193].

Resistance to chemotherapy and targeted therapies is a major problem in current leukemia treatment [334]. Therefore, nanotechnology has become one of the focal points of the modern medical research that looks for improving the performance of cancer treatment [335]. Research focused on leukemia therapies using green biogenic MNPs has reported that these new nanosystems could support leukemia therapy and diagnosis [324]. Leukemia cells are not initially reflected as a solid tumor, which could be of greater relevance in understanding this pathology, highlighting the potential protection of healthy cells [336]. NPs have provided innovative non-invasive and straightforward platforms to diagnose and treat liquid tumors. In this sense, the protection of healthy cells by the antioxidant and antitumoral properties of the MNPs generated by biological synthesis is crucial for leukemia therapy. In addition, functionalized delivery systems could positively impact cases with drug resistance [334,337,338,339,340]. MNPs inhibit cell proliferation and clonogenesis, induce apoptosis, and may cause cell cycle arrest, mechanisms that significantly reduce the IC_50_ values of conventional drugs. Some reports mention side effects, possibly due to ROS-dependent up-regulation of leukemic cells. These biogenic NPs aim to reduce systemic toxicity by protecting healthy cells due to their antioxidant and antitumoral properties [334,335,341]. Table 11 shows some in vitro studies of Au and AgNPs developed to improve the treatment of leukemia.

## 5. Discussion

In recent years, the development of greener methods to synthesize AuNPs and AgNPs has shown significant scientific and technological success. It is an area of great interest and presents excellent growth potential. Green biosynthesis is currently carried out using extracts of natural products, such as plant leaves, fruits, peels, derived products and waste, or microorganisms. These natural sources and procedures provide several advantages over classical chemical and physical methods since they are environmentally friendly and cost effective. Several AgNPs and AuNPs have been biosynthesized and evaluated due to their potential application against cancer. They stand out as ideal nanosystems for addressing current and forthcoming issues in cancer treatment. The robustness of gold and silver as building blocks to engineer novel functional biomaterials for applications in medicine is due to their excellent bacteriostatic anticorrosive and antioxidative properties. Significant advances in preparing these smart nanodevices have been made by either chemical or biological approaches. Over recent years, green methods to synthesize MNPs for cancer therapy have largely evolved because of their facile handling in the laboratory. The starting materials are relatively more affordable than those required in chemical synthesis. In addition, from an environmental point of view, the preparation of MNPs mediated by biological intermediates has a positive impact since it makes possible the use of water-based, rather than solvent-based, systems, which generate huge amounts of toxic waste each year.

Ag and AuNPs are particularly appealing due to their tunable surface chemistry, which allows the conjugation of a plethora of functional groups for specific tasks. Herein, we have shown that thiol linkers can be used to attach a wide range of ligands to MNPs. Therefore, the selection of a proper molecule (e.g., a small-organic compound), macromolecule (e.g., oligomers or polymers), or even therapeutic agents, DNA, amino acid, protein, peptide, or oligonucleotides may be used to decorate the surface of MNPs. Thus, NP functionalization and bioconjugation increase treatment efficacy, limiting off-target toxicity. In addition, by selecting a suitable ligand, it is possible to design stimuli-responsive nanodevices that, once accumulated in the tumor site, can be activated either intrinsically or extrinsically. For instance, extracellular pH in tumors is slightly more acidic than normal cells. Indeed, this difference in the TME is exploited to design effective pH-responsive MNPs, which can provide significant usefulness in the controlled release of bioactive cargoes.

However, given that the starting biological material is complex in nature, the formed MNPs are not always fully characterized. In addition, it is difficult to discern between the damage triggered by the biological molecules or MNPs. In this sense, batch-to-batch reproducibility may also be compromised. Indeed, biosynthesized green MNPs have tried to find a place in the field of cancer therapy. They have been extensively evaluated in different cancer cell lines grown as monolayers. Although these evaluations have made it possible to determine the cytotoxicity induced by MNPs and determine some underlying mechanisms, these findings are not easily extrapolated to a whole organism. Three-dimensional cancer models offer a higher level of complexity, so they could be more accurate in decreasing the gap between conventional culture studies and in vivo responses. Surprisingly, they have been scarcely used to evaluate biogenic MNPs, and only a few green MNPs have been examined for their antitumor activity in vivo. Although it is desirable that research moves to in vivo models to extend in vitro findings, it is also predictable that this transition will face novel challenges, such as biosafety, biocompatibility, physiological effects, stability, biodistribution, circulation time, and specificity of biosynthesized MNPs.

Despite all the therapies developed for cancer therapy, it continues to be an incurable disease and increases over time. Currently, a targeted therapy using MNPs is an excellent opportunity to effectively treat cancer since NPs have shown enormous potential in different cancer types such as all those presented in this review. Nevertheless, more studies must be carried out. It is very important to encourage collaborative work to better elucidate the properties and potential application of green biosynthesized NPs in cancer therapy. This requires enthusiastic chemists, biologists, pharmacists, physicians, and physicists to understand the in vitro and in vivo mechanism of action of AgNPs and AuNPs. Such efforts will surely enhance the safe consumption of NPs in the pharmaceutical industry to design, evaluate, and make contributions for customized nanotherapeutic medicines. AuNPs and AgNPs obtained by green synthesis are not extensively used in drug delivery. It could be interesting to accelerate this research to eliminate concerns in terms of their toxicity, stability, behavior, absorption, distribution, metabolism, and excretion. In addition, long-term studies of AgNPs and AuNPs in vivo are necessary to evaluate toxicity and performance.

## 6. Conclusions

The use of MNPs as drug delivery systems is particularly attractive for both the development of new strategies and the optimization of existing treatments against cancer. The green biosynthesis of AuNPs and AgNPs can be a good option to provide NPs with defined sizes, optimal morphologies, and good stability. However, the use of plant extracts to decorate the core of MNPs has been little studied. The undefined mixture of compounds in the natural extracts may be an obstacle to the functionalization of MNPs. The presence of various functional groups can hinder the selectivity and interaction of the MNPs with a cellular receptor. It is important to identify the main active principle of the herbal extract, which provides a way to detect the chemical interaction between functional groups, both with the MNP core itself and with the target. In addition, green nanoparticles can be used to combine different anticancer techniques, e.g., magnetic hyperthermia when made of superparamagnetic iron oxide NPs, PTT when AuNPs are used, and PDT or gene silencing by siRNA delivery. Moreover, green nanoparticles cannot only be used for therapy purposes but also for diagnostics (theragnostics), for instance, by doping the inner core with gadolinium, iron oxide, or manganese acting as MRI contrast agents. Although there are many studies on the action of MNPs against the biological systems, most of them have been carried out in vitro, where only effects such as necrosis, apoptosis, or ROS generation have been studied, but deep knowledge on pharmacokinetics, metabolism, accumulation, and distribution in different parts of the organism is still lacking. AgNPs and AuNPs combined with antineoplastic drugs have demonstrated their potential pharmacological effect; thus, it is important to highlight the necessity of further studies using 3D cell cultures and ex-vivo and in vivo models to explore the complex multicellular TME, providing essential and integrative information required for advanced cancer therapy.

## Figures and Tables

**Figure 1 pharmaceutics-13-01719-f001:**
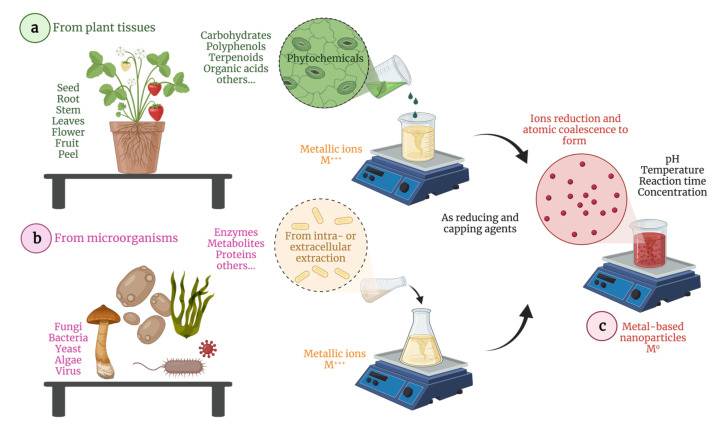
Schematic representation of the biosynthesis of MNPs (**c**) mediated by plants tissues (**a**) and microorganism derivatives (**b**). Illustration created with BioRender.com (accessed on 22 August 2021).

**Figure 2 pharmaceutics-13-01719-f002:**
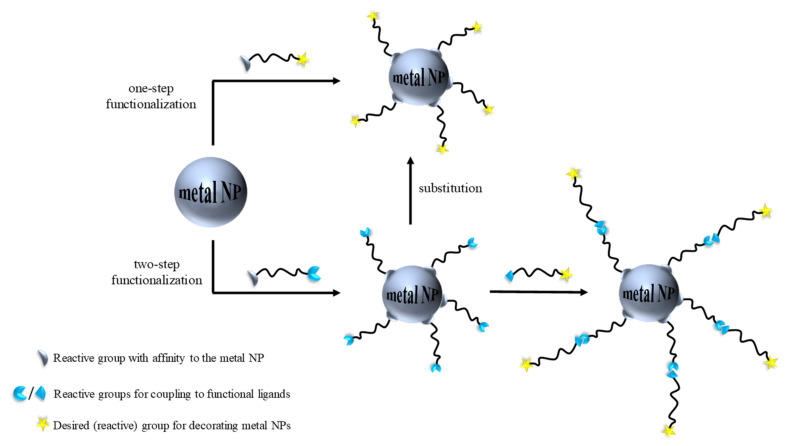
Two approaches are instrumental in producing decorated MNPs by utilizing appropriate functional ligands in either one-step or two-step processes.

**Figure 3 pharmaceutics-13-01719-f003:**
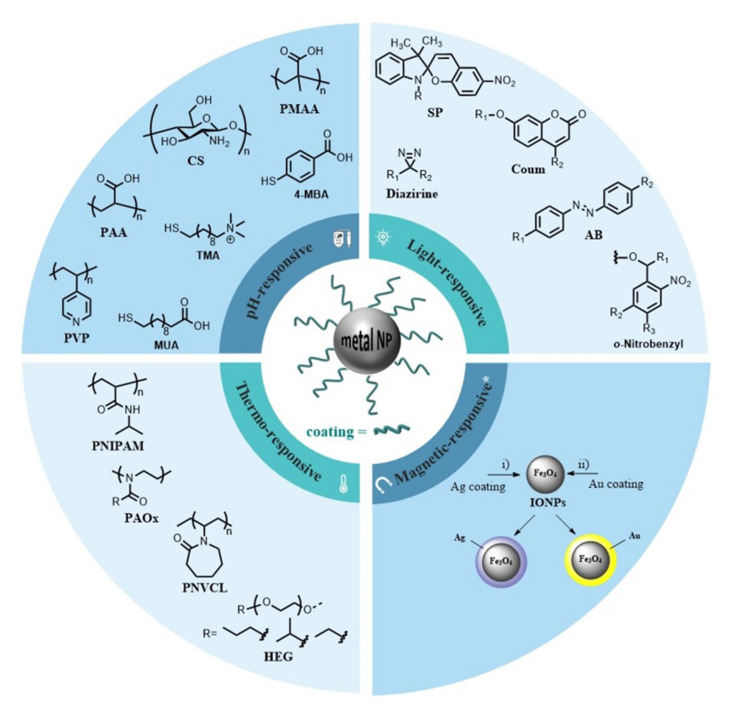
Molecules and macromolecules frequently used to coat MNPs resulting in stimuli-sensitive nanosystems. * Magnetic core NPs (i.e., Fe3O4 NPs) are decorated with silver or gold to obtain magnetic-responsive nanosystems.

**Figure 4 pharmaceutics-13-01719-f004:**
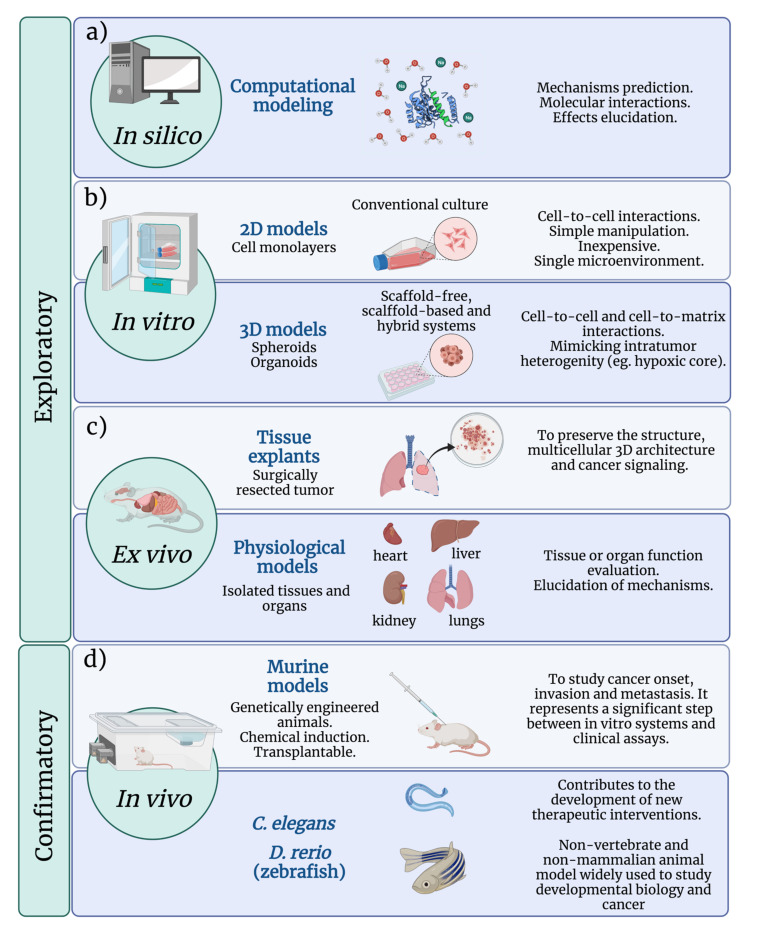
Features of (**a**) in silico (**b**) in vitro, (**c**) ex vivo, and (**d**) in vivo MNP evaluation models.

**Figure 5 pharmaceutics-13-01719-f005:**
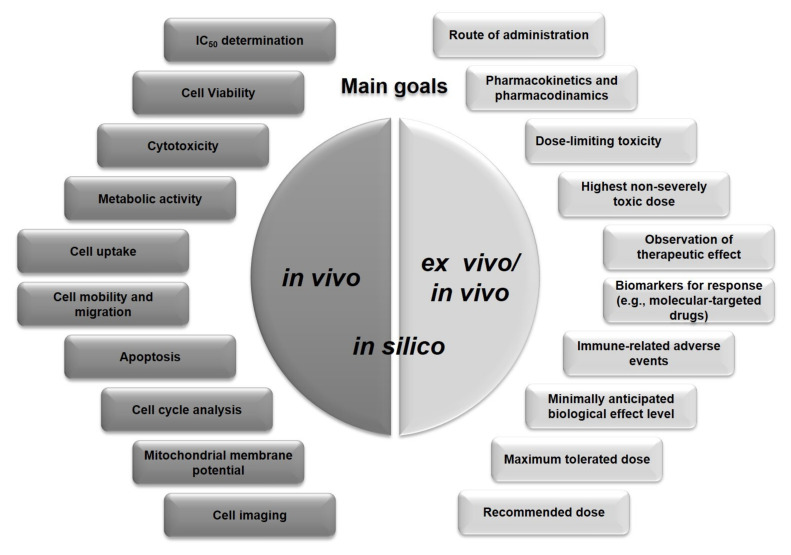
Main goals for using in silico, in vitro, ex vivo, and in vivo models for MNP evaluation.

**Table 1 pharmaceutics-13-01719-t001:** pH-sensitive chemical bonds and release mechanisms in acidic conditions. Adapted from [55], published by MDPI, 2020.

pH-Sensitive Bonds	Chemical Mechanisms
Imine	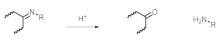
Hydrazone	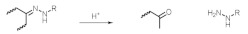
Oxime	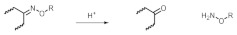
Amide	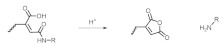
Acetals	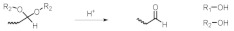
Orthoester	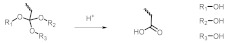

**Table 2 pharmaceutics-13-01719-t002:** Evaluation of AuNPs and AgNPs in *C. elegans* and Zebrafish; type of NP, concentration, specimen, stage of development, transcriptome profiling method, up or down regulation, and gene ontology.

Type of NP, Concentration and Specimen	Stage of Development at RNA Extraction and Method of Transcriptome Profiling	Up- or Down-Regulation	Gene Ontology	Reference
Citrate-coated AuNPs 5.9 mg/L *C. elegans* L3 larval stage exposed for 12 h.	L3 larval stage Affymetrix *C. elegans* gene chip	Up-regulation	Amyloid Processing Citrate cycle Clathrin-mediated endocytosis Apoptosis Unfolded protein response G-protein signalling	[171]
Bare AuNPs 0.1 and 0.2 mg/L MUA/Au = 0.5 MUA/Au = 3 *C. elegans* embryos exposed for 72 h.	Adults Affymetrix *C. elegans* gene chip	Up-regulation Down-regulation	Defense response Lipid catabolic processes Lipid storage Body morphogenesis Body shape and size regulation Metal detoxification(homeostasis) and stress Lifespan	[172]
Citrate-capped AuNPs 20 mg/L Liver of adult Zebrafish fish exposed for 96 h.	Adults Affymetrix GeneChip Zebrafish Genome Array	Up-regulation Down-regulation	Stress response Development The establishment of localization Biogenesis Metabolic process Locomotion Biological adhesion Response to a stimulus	[173]
AgNPs 10 μg/mL *C. elegans* L1 larvae trough adulthood exposed for 72 h.	Adults RNA seq/*C. elegans*	Up-regulation Down-regulation	Cellular process: cell cycle, meiosis, apoptosis, etc. Growth Metabolic process Reproduction Cellular component organization and biogenesis Behavior Cellular process: cell communication, cell surface receptor signaling pathway Developmental process. Morphogenesis, generation of neurons Metabolic process Response to stimulus Growth Locomotion Reproduction	[174]
AgNPs 50 μg/L Gills from adult zebrafish exposed for 28 days	Adults Agilent Technologies 4x44 K zebrafish microarray	Over-represented ND	Extracellular components (Matrix) Mithocondria Ribonucleoprotein complexes DNA damage DNA/RNA processing Heat shock proteins Cell growth and migration Anatomical, organ and cell morphogenesis Embryonic, skeletal and organ development	[175]
Bare AgNPs 0.01 mg/L 6 dpf larvae exposed for 15 days	Larvae Agilent’s zebrafish (V3) oligonucleotide microarrays	Up-regulation Down-regulation	Photoreception Circadian clock regulation Cardiovascular disease Genetic disorder Hematological disease Hypersensitivity response Ophthalmic disease	[176]
Citrate-coated AgNPs 0.4 mg/L 24 hpf embryos exposed for 24 h	Embryos Agilent Technologies 4x44 K zebrafish microarray	Down-regulation	Ligans-gated ion channel activity Dopaminergic receptor signaling and neuro differentiation Neuron recognition, Regulation of neurogenesis	[177]

**Table 3 pharmaceutics-13-01719-t003:** Advantages and disadvantages of different murine models used cancer research. Adapted from [166], published by The Company of Biologists, 2017.

Model	Way of Generation	Advantages	Disadvantages
Ectopic CDX Cell line-Derived Xenograft	Human tumor cells (fluorescents or not) Implanted subcutaneously	Easy, fast, and cheap Commercial cell lines or primary cell cultures Eeasily measurable	Immunodeficient host Some cancer types fail to grow Not specific tissue growing
Orthotopic CDX Cell line-Derived Xenograft	Implantation on specific tissue	Microenvironment similar to the origin of the tumor Eeasily measurable	They are more technically complex than ectopic Immunodeficient host Not all cancer types grow
Metastatic CDX	By injection of tumor cells by vein or intra-cardiac	Tumors can grow in a variety of tissues or organs	The model does not mimic the original tumor Technically demanding to detect the location of tumor
PDXPatient-Derived Xenografts	By implantation of tumor cells or fragment derived from human tumors. (ectopically or orthotopically)	Tumors generated maintain the phenotypic and genotypic characteristics of the original tumor derived from the patient	Requires fresh patient tumor tissue Immunodeficient host Relatively expensive Slow implementation Technically demanding
Syngeneic	Mouse tumor tissue or cells implanted on same strain mouse	Good growing tumors Microenvironment adequate Immunocompetent host	Mouse microenvironment Not useful as human model
Conventional GEMM Genetically Modified Mouse Models	Oncogenic-driven transgenic mice to develop specific cancer.	Natural microenvironment Intact immune system Modelling of early/late stages of tumor progression	Mouse microenvironment Not truly of human disease Not valuable for certain test therapies

**Table 4 pharmaceutics-13-01719-t004:** Murine in vivo models used to evaluate the antitumor activity of green biogenic AuNPs and AgNPs.

In Vivo Model	Biological Source	Type of NP	Size (nm) and Shape	Cell Line	Administration Route	Reference
Subcutaneous Breast cancer Mice	*Curcuma longa*	AuNPs	278 Spherical	DMBA	Oral	[185]
Breast cancer Mice	*Mangifera indica*	AuNPs	55.5–65.5 Spherical	MDA-MB-231	Oral	[186]
Subcutaneous Cacu Mice	Peptides	AuNPs	100–150 Spherical Conjugate	HeLa	Tail vein	[187]
Subcutaneous Lung cancer Mice	*Dimocarpus longan*	AgNPs	13 Spherical	H1299	Intraperitoneal	[188]
Intravenous Mice	*Curcumin*	AgNPs	50–100	HeLa	Intraperitoneal	[189]
Mice	*Nostoc carneum*	AgNPs	16 Spherical	EAC	Intraperitoneal	[190]
Subcutaneous C57 Mice	*Curcumin*	AuNPs	16 Spherical	C540 (B16/F10)	Intratumoral	[191]
Intravenous Mice	*Camellia sinensis*	AuNPs	20–30 Spherical	HL-60	Intravenous	[192]

**Table 5 pharmaceutics-13-01719-t005:** Evaluation of green biogenic Au and AgNPs in skin cancer cells; biological source, NPs characteristics, IC_50_, and cell line.

Biological Source or Hybridization	Type of NP	Size (nm)	Shape	IC_50_ (µg/mL)	Cell Line/In Vivo Model	Reference
*Siberian ginseng*	AuNPs	200	Spherical	10 µg/mL	B16	[202]
Fluorouracil	AuNPs	16–150	Spherical	ND	A431	[204]
TiO_2_	AuNPs	148–333	Spherical	ND	B16F1	[208]
*Bacillus licheniformis*	AgNPs	20–80	Triangular	2457.5	B16F10, A431	[209]
TiO_2_	AgNPs	50	Nanoprism	100	B16F10, C57BL/6J mice	[122]
*A. muricata*	AgNPs	50	Spherical	ND	A375	[203]
BSA	AgNPs	58	Spherical	200	B16F10	[206]

ND: not determined.

**Table 6 pharmaceutics-13-01719-t006:** Evaluation of green biogenic Au and AgNPs in breast cancer cells; biological source, NPs characteristics, IC_50_, and cell line.

Biological Source	Type of NP	Size (nm)	Shape	IC_50_ (µg/mL)	Cell Line	Reference
*Actinobacterial-SF23*	AgNPs	3–36	Spherical	16.30	MCF7	[218]
*Buchanania axillaris*	AgNPs	17–80	Spherical	31.20	MCF7	[219]
*Black Tea mistry*	AgNPs	9–15	Spherical	30	MCF7	[220]
*Oscillatoria limnetica*	AgNPs	3–17	Spherical	6.14	MCF7	[221]
*Dunaliella salina*	AuNPs	~22	Spherical	98	MCF7	[222]
*Fagonia indica*	AgNPs	10–60	Spherical	12.35	MCFT	[223]
*Linum usitatissimum*	AuNPs	~31	Triangular	5	MCF7	[224]
Chitosan-functionalized	AgNPs	13–22	Spherical	6.40 6.56	MCF7 MDA-MB-231	[225]
*Polysiphonia algae*	AgNPs	5–25	Spherical	4.19	MCF7	[226]
*Cynara scolymus*	AgNPs	98	Spherical	10	MDA-MB-231	[227]
*Syzygium jambolanum*	AgNPs	20–25	Spherical	ND	MCF7	[228]
*Tamarindus indica*	AgNPs	20–52	Spherical	20	MCF7	[229]
*Commiphora wightii*	AuNPs	~28	Spherical	66.11	MCF7	[230]
*Sargassum ilicifolium*	AuNPs	20–25	Spherical	24	MDA-MB-231	[231]
*Mentha longifolia*	AuNPs	~36	Spherical	274	MCF7	[232]
*Acacia luciana*	AgNPs	50	Spherical	4.37	MCF7	[233]
*Garcinia atroviridis*	AgNPs	5–30	Spherical	2.00	MCF7	[234]
*Allium saticum*	AgNPs	20–35	Spherical	89.86	MCF7	[235]
*Cladosporium sp*	AuNPs	5–10	Spherical	38.23	MCF7	[236]
*Curcuma mangga*	AuNPs	~28	Spherical	0.41	MCF7	[237]
*Dragon fruit*	AuNPs	10–20	Spherical	ND	MCF7 MDA-MB-231	[238]

ND: not determined.

**Table 7 pharmaceutics-13-01719-t007:** Evaluation of green biogenic Au and AgNPs in lung cancer cells (A549 cell line); biological source, NPs characteristics, and IC_50_.

Biological Source	Type of NP	Size (nm)	Shape	IC_50_ (µg/mL)	Cell Line	Reference
*Rabdosia rubescens*	AuNPs	130	Spherical	25	A549	[242]
*Millettia pinnata*	AuNPs	37	Spherical	14.76	A549	[245]
*Pinus roxburghii*	AgNPs	80	Spherical	11.28	A549	[243]
*Pleuropterus multiflorus*	AgNPs AuNPs	~275 ~105	Spherical	35.16	A549	[244]
*Matricaria chamomilla*	AgNPs	~45	Spherical	62.82	A549	[246]
*Carpesium cernuum*	AgNPs	13	Spherical	ND	A549	[105]
*Marsdenia tenacissima*	AuNPs	50	Spherical	15	A549	[247]
*Beta vulgaris*	AgNPs	5–20	Spherical	48.20	A549	[248]
*Garcinia mangostana*	AgNPs AuNPs	13–31 15–44	Asymmetric dumbbell Spherical	ND	A549	[249]
*Cratoxylum formosum Mucuna birdwoodiana*	AgNPs	~9 ~35	Spherical	ND	A549	[249]
*Musa paradisiaca*	AuNPs	50	Spherical to triangular	58	A549	[250]
*Dendropanax morbifera*	AgNPs AuNPs	100–150 10–20	Polygonal Hexagonal	ND	A549	[251]
*Cymbopogon citratus*	AgNPs	17–26	Spherical	25	A549	[252]
*Indigofera tinctoria*	AgNPs AuNPs	~19	Spherical	56.62 59.33	A549	[253]
*Coptis chinensis*	AgNPs	6–45	Spherical	15	A549	[254]
*Magnolia officinalis*	AuNPs	128	Spherical	~18	A549	[255]
Garlic, Green tea, Turmeric	AgNPs	8	Spherical	13.26 17.25 11.11	A549	[256]
*Rosa damascena*	AuNPs	8–45	Spherical to triangular	ND	A549	[257]
*Nannochloropsis sp.*	AgNPs	~57	Spherical	15	A549	[258]
*Borago officinalis*	AgNPs	30–80	Spherical Hexagonal	~7	A549	[259]
*Moringa oleifera*	AuNPs	10–20	Spherical	98.46	A549	[260]
*Euphrasia officinalis*	AgNPs AuNPs	~40 ~50	Quasi-spherical	~2 ND	A549	[261]
*Curcumae kwangsiensis*	AgNPs	15–21	Spherical	249 187 152	HLC-1 LC-2/ad PC-14	[262]
*Escherichia coli VM1*	AgNPs	10–15	Spherical	40	A549	[263]

ND: Not determined.

**Table 8 pharmaceutics-13-01719-t008:** Evaluation of green biogenic MNPs in prostate cancer cells; biological source, NPs characteristics, IC_50_, and in vitro models.

Biological Source	Type of NP	Size (nm)	Shape	IC_50_ µg/mL	Cell Line	Reference
*Cyclopia intermedia*	AuNPs	20	Spherical and triangular	ND	PC-3	[272]
*Acai berry*	AuNPs	172	Spherical and triangular	1000	PC-3	[273]
Casein hydrolytic peptide	AuNPs	~20	Hexagonal	ND	DU-145	[274]
Doxorubicin	AuNPs	~75	Spherical	ND	LN-CaP	[275]
*Piper Nigrum*	AgNPs	15-38	Spherical	ND	PC3	[276]
*Salvia miltiorrhiza*	AgNPs	100	Spherical, oval, hexagonal, and triangular	ND	LNCaP	[277]
*Piperlongum*	AgNPs	5–35	Quasi-spherical	73.41 38.53	PC-3 DU-145	[278]
*Salacia chinensis*	AgNPs	40–80	Rods, triangular and hexagonal	7.46	PC-3	[279]
*Saraca asoca*	AgNPs	36	Spherical	50.00	DU-145	[280]
*Cornus officinal*	AgNPs	~12	Quasi-spherical	25.54	PC-3	[281]
*Eclipta prostrata*	AgNPs	50–75	Spherical	9.84	PC-3	[282]
*Moringa oleifera*	AgNPs	44–60	Spherical	25.21	PC-3	[282]
*Thespesia populnea*	AgNPs	47–97	Spherical	31.21	PC-3	[282]
*Guiera senegalensis*	AgNPs	50	Spherical	23.48	PC-3	[283]
*Pestalotiopsis microspora*	AgNPs	2–10	Spherical	27.71	PC-3	[284]
*Dimocarpus Longan Lour*	AgNPs	9–32	Spherical	<10.00	PC-3	[285]
*Plumbago zeylanica*	AgNPs	80–98	Spherical and cuboid	58.61	PC-3	[286]
*Semecarpus anacardium*	AgNPs	60–95	Spherical and cuboid	42.77	PC-3	[286]
*Terminalia arjuna*	AgNPs	34–70	Spherical and cuboid	41.78	PC-3	[286]
*Gracilaria edulis*	AgNPs	55–99	Spherical	39.60	PC-3	[287]
*Alternanthera sessilis*	AgNPs	30–50	Spherical	6.85	PC-3	[288]
*Cell-free supernatant of actinobacteria*	FeNPs	65–87	Spherical	65	PC-3	[289]
*Rhus punjabensis*	FeNPs	~48	Spherical	12.79	DU-145	[290]
*Leucaena leucocephala*	ZnNPs	50–200	spherical	103.72	PC-3	[291]
*Cinnamomum tamala*	TiNPs	23	Irregular	ND	DU-145	[292]

ND: not determined.

**Table 9 pharmaceutics-13-01719-t009:** Evaluation of green biogenic Au and AgNPs in colorectal cancer; biological source, NPs characteristics, IC_50_, and in vitro models.

Biological Source	Type of NP	Size (nm)	Shape	IC_50_ (µg/mL)	Cell Line	Reference
*Zingiber officinale*	AgNPs	42–61	Spherical	150.80	HT29	[295]
*Albizia lebbeck*	AuNPs	20–30	Spherical	48	HCT-116	[297]
*Pleurotus sajor-caju*	AgNPs AuNPs	16–18 4–22	Spherical	50 80	HCT-116	[298]
*Trichosanthes kirilowii*	AuNPs	50	Spherical	ND	HCT-116	[301]
*Aspergillus niger*	AgNPs	20–25	Spherical	160	HT-29	[302]
*Anthemis atropatana*	AgNPs	38.89	Spherical	4.88	HT29	[303]
*Chaetomorpha linum*	AgNPs	35	Spherical	48.84	HCT-116	[304]
*Bergenia ciliata*	AgNPs	50–100	Spherical	ND	HT29	[305]
*Albizia lebbeck*	AuNPs	20–30	Spherical	48	HCT116	[297]
*Abutilon indicum*	AuNPs	1–20	Spherical	210	HT-29	[306]
*Nostoc sp.*	AgNPs	14.9	Spherical	150	Caco2	[307]
*Annona muricata*	AgNPs	16–20	Spherical	ND	HCT116	[203]
*Mentha arvensis*	AgNPs	12–40	Spherical triangular	1.7	HCT116	[300]
*Perilla frutescens*	AgNPs	~26	Spherical rhombic triangle	39.28	Colo205	[308]
*Ulva lactuca* L.	AuNPs	6–20	Spherical	98.46	HT29 Caco-2	[260]
*Curcuma longa Zingiber officinale*	AgNPs	20–51	Quasi-spherical	150.80	HT29	[295]
*Artemisa tournefortiana*	AgNPs	22	Spherical	40.71	HT29	[309]
*Wedelia trilobata*	AuNPs	10–50	Spherical	ND	HCT15	[310]

ND: not determined.

**Table 10 pharmaceutics-13-01719-t010:** Evaluation of green biogenic Au and AgNPs in HeLa and SiHa cells of cervix adenocarcinoma; biological source, NPs characteristics, and IC_50_.

Source of Nanoparticles	Type of NP	Size [nm]	Shape	IC_50_ (µg/mL)	Cell Lines	Authors
*Allium saralicum*	AgNPs	20–40	Spherical	>1000	HeLa	[318]
Lycopene	AgNPs	50–100	Spherical	ND	HeLa	[296]
*Mangifera indica*	AgNPs	9–61		~400	HeLa	[319]
*Nepeta deflersiana*	AgNPs	33	Spherical	~3.9	HeLa	[320]
*Punica granatum*	AgNPs	41–69	Spherical	ND	HeLa	[321]
*Styrax benzoin*	AgNPs	12–38	Spherical	ND	HeLa	[322]
*Ziziphus jujube * + graphene oxide	AgNPs	26 7	Spherical	ND	HeLa	[323]
Curcumine derivative (ST06)	AgNPs	50–100	Spherical	1 µM	HeLa	[189]
*Ginkgo biloba*	AgNPs	40	Spherical	4 (HeLa) 6 (SiHa)	HeLa SiHa	[324]
*Leucas aspera*	AgNPs	35–54	Spherical	36	HeLa	[325]
*Taxus baccata*	AgNPs	75–91	Spherical	10 µg/mL	HeLa	[326]
*Alternanthera sessilis*	AuNPs	30–50	Spherical	>5 µg/mL	HeLa	[327]
*Benincasa hispida*	AuNPs	22	Spherical	2.3 µg/mL	HeLa	[328]
*Catharanthus roseus*	AuNPs	25–35	Spherical	5 µg/mL	HeLa	[329]
*Celastrus hindsii*	AuNPs	13–53	Spherical	12.5	HeLa	[330]
*Zataria multiflora*	AuNPs	42	Spherical	ND	HeLa	[331]

ND: not determined.

**Table 11 pharmaceutics-13-01719-t011:** Evaluation of green biogenic Au and AgNPs in leukemia cells; biological source, NPs characteristics, IC_50_, and in vitro models.

Biological Source	Type of NP	Size (nm)	Shape	IC_50_ (µg/mL)	Cell Line	Reference
*Glycyrrhiza glabra L.* Extract	AgNPs	20	Spherical	604 467 445 438	J45.01, J.Clone E6–1, J.CaM1.6, J.RT3-T3.5	[342]
*Sargassum muticum* water Extract	AuNPs	10	Spherical	4.22 5.71 6.55 7.29	K562, HL-60, Jurkat, CEM	[343]
*Hibiscus sabdariffa * Extract	AuNPs	15–45	Spherical	761 803 882	C1498, Human HL, 32D-FLT3-ITD	[344]
*Thymus vulgaris*	AuNPs	10–30	Spherical	397	C1498, Human HL, 32D-FLT3-ITD	[345]
*Camellia sinensis * Tea	PdNPs	6–18	Spherical	ND	MOLT-4	[346]
*Achillea millefolium*	AgNPs	~22	Spherical	0.011	MOLT-4	[347]
Archaeoglobus fulgidus chimeric ferritin	AgNPs	4.5	Spherical	ND	NB4	[348]
*Verbena officinalis* Extract	Au/CuO/ZnONPs	35	Spherical	0.64 µmol	Jurkat	[349]
*Glechoma hederacea L.* Extract	Au/CuO/ZnONPs	10	Spherical	ND	Jurkat	[350]
Cyanobacterial strains *Leptolyngbya tenuis*, *Coleofasciculus chthonoplastes*, and *Nostoc ellipsosporum*	AuNPs	8–42	Spherical	150	MOLT-4 T-ALL	[351]

ND: not determined.

## Data Availability

The review is based on published data and the sources of data upon which conclusions have been drawn can be found in the reference list.

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
