# Peer review of "Green Metallic Nanoparticles for Cancer Therapy: Evaluation Models and Cancer Applications"

_pharmaceutics, 2021, doi:10.3390/pharmaceutics13101719_

Round 1

Reviewer 1 Report

In general, the manuscript is not well organized, so it is very difficult to follow. I will reconsider it after major corrections and improvements of the first version. 

Major comments

  • At the end of the “abstract”, you must present concluding remarks from your study, while you mentioned about the structure of the paper without no description about your conclusion.
  • It is better to have a short introduction section without subsections. Please add a three or four paragraph about the introduction. Then, rearrange the other parts.
  • In general, the manuscript seems a little bit shallow. What are the main contributions of this paper compare to previously published ones? It is not clear for me.
  • Some important recently published papers are missed. You can add some of them as: 1016/j.nano.2015.07.015 & 10.1016/j.jconrel.2020.08.012 & 10.1063/1.5052455
  • More details need to be considered in different sections. Additionally, you must improve the captions of the figures so that they independently express the concept.
  • What is your idea about adding “in silico analysis”? It is an important part of research like in-vitro and ex-vivo, it is not included in this study. My suggestion is that you must add this viewpoint to your study. For more information, please refer to 1016/j.nantod.2020.101057 10.3389/fonc.2021.655781 and 10.3390/cancers13102481.
  • Maybe some comprehensive figures or tables can be helpful to reduce the number of words, while it is more straight-forward and clear. Apply this wherever you can in the new version.
  • In section 2.4.1, you have a lot of small paragraphs (4-5 line). Please combine them to a large enough paragraph that follows a similar idea. Apply this to the whole manuscript.
  • Manuscript lacks of “Discussion” section. Add a discussion about the results of this review paper for presenting a guideline for future studies in the field. This is the most important part of a review paper.
  • The "Conclusions" section is very weak. Rewrite it.

Typo and grammatical comments:

A better English grammar editing must be applied to improve the level of presenting. Please double-check all the manuscript. I give some clue about them as:

  • In the keywords are as follows: “Keywords: nanosytems 1; nanoparticles 2; cancer treatment; 3; in vitro model 4; ex 52 vivo model 5; 6; in vivo model.)”. There is a problem in numbers!
  • In lines 145, 147, 151, you used MNPs. Then in line 158, you defined metal-based nanoparticles as MNPs. Correct it and also other similar issues.
  • The title of sub-section in line 624 is “Magnetic”. What do you mean? Magnetic NPs or Magnetic groups?
  • The title of subsection in line 1387 is as:” Applications of green Smart nanosystems in cancer therapy”, why the first letter of “smart” is capital? Please check the whole manuscript on this issue?
  • In table 9, “nd” is not defined.
  • “Reference” section is repeated!

Reviewer 2 Report

The present review manuscript entitled “Green metallic nanosystems for cancer therapy: evaluation models and cancer applications” by Tinajero et al involves the analyses of smart metallic systems as delivery systems of bioactive molecules for cancer therapy. The present important review will complement currently published information such as:

Andleeb et al. A Systematic Review of Biosynthesized Metallic Nanoparticles as a Promising Anti-Cancer-Strategy. Cancers 2021, 13, 2818. https://doi.org/10.3390/cancers13112818

Moreover, this interesting review focuses on the green metallic nanosystems: their obtention, the in vitro, ex vivo, and in vivo models used for evaluating these systems, and the applications of then in different types of cancer.

The manuscript has a complete and well-organized introduction, specified methodologies according to the explained models and diversity used of tables and images in order to show the summarize information. Additionally, the manuscript is clear and easy to follow according to the development of this topic.

I encourage the authors to check some typographical mistakes (pdf attached, in red).

Line 51: abstract. Two final points at the end.

Lines 52 and 53: keywords without the numbers and the final parenthesis

Line 101: sp2 should be sp2 (superscript)

Lines 199, 219, 460, 560, etc.; please use the reference without the year and the first initial. As an example, in line 199: Ranoszek-Soliwoda et al

Lines 349, 352, 378: there is one extra blank space between words.

Line 365: it should be “among others”

Line 427: nucleic acids

Line 659: XXX cells? (Please describe what XXX means)

Line 1305: Tabla should be Table.

Moreover, it will be important to summarize the in vitro methods because they are normal methodologies for these kinds of assays. As an example, the cell viability assay (MTT, MTS, WST-8, TB) is standard and it is not necessary to describe the reaction, wavelength information, etc. As a suggestion the references for these methods should be enough for the reader.

The authors would summarize all the information in one table in order to improve this part.

Furthermore, the size of the tables and the image quality should keep the  requirements of the editorial (tables are out of the margins and images looks pixelated).

Round 2

Reviewer 1 Report

The manuscript are improved significantly and it is ready to publication.